# Bispectra of climate cycles show how ice ages are fuelled

Diederik Liebrand[1], Anouk T. M. de Bakker[2, 3]

[1]Center for Marine Environmental Sciences (MARUM), University of Bremen, Bremen, Germany
[2]LIttoral ENvironnement et Sociétés (LIENSs), Université de La Rochelle, La Rochelle, France
[3]Unit of Marine and Coastal Systems, Deltares, Delft, the Netherlands

*Correspondence to*: D. Liebrand (diederik@palaeoclimate.science), A. T. M. de Bakker (Anouk.deBakker@deltares.nl)

**Abstract.** The increasingly nonlinear response of the climate-cryosphere system to insolation forcing during the Pliocene and Pleistocene, as recorded in benthic foraminiferal stable oxygen isotope ratios ($\delta^{18}O$), is marked by a distinct evolution in
ice-age cycle frequency, amplitude, phase, and geometry. To date, very few studies have thoroughly investigated the nonsinusoidal shape of these climate cycles, leaving precious information unused to further unravel the complex dynamics of the Earth's system. Here, we present higher-order spectral analyses of the LR04 $\delta^{18}O$ stack that describe coupling and energy exchanges among astronomically-paced climate cycles. These advanced bispectral computations show how energy is passed from precession-paced to obliquity-paced climate cycles during the Early Pleistocene (~2,500–~750 ka), and ultimately to
eccentricity-paced climate cycles during the Middle and Late Pleistocene (from ~750 ka onward). They also show how energy is transferred among many periodicities that have no primary astronomical origin. We hypothesize that the change of obliquity-paced climate cycles during the mid-Pleistocene transition (~1,200–~600 ka), from being a net sink into a net source of energy, is indicative of the passing of a land-ice mass-loading threshold in the Northern Hemisphere (NH), after which cycles of crustal depression and rebound started to resonate with the ~110-kyr eccentricity modulation of precession.
However, precession-paced climate cycles remain persistent energy providers throughout the Late Pliocene and Pleistocene, which is supportive of a dominant and continuous fuelling of the NH ice ages by insolation in the (sub-) tropical zones, and the control it exerts on meridional heat and moisture transport through atmospheric and oceanic circulation.

## 1. Introduction

The recurrent ice ages of the Pliocene and Pleistocene, as captured in benthic foraminiferal $\delta^{18}O$ records, are characterized by long-term trends in glacial-interglacial cycle duration, amplitude, response time, and geometry (Fig. 1 and Fig. 2). These trends mainly reflect the increasingly nonlinear response of the (northern) high latitude cryosphere and global deep-sea temperatures to radiative forcing, i.e. the combined greenhouse effect of incoming solar radiation (insolation) and the partial pressure of atmospheric $CO_2$ ($P_{CO_2^{atm}}$), when global climatic conditions deteriorated (Lisiecki and Raymo, 2005, 2007;
Martinez-Boti et al., 2015; Chalk et al., 2017). Spectral analysis, in combination with age control independent from astronomical tuning, formed the decisive evidence in support of Milankovitch's theory of astronomical climate forcing (Broecker, 1966; Hays et al., 1976). These statistical methods, such as the (time-evolutive) fast Fourier transform, or wavelet

analysis using a Morlet-transform, perform well in defining sinusoidal cycle properties, such as frequency, amplitude, and/or (cross-) phase relationships with respect to an astronomical reference curve (Lourens and Hilgen, 1997; Lisiecki, 2010; Meyers and Hinnov, 2010). However, spectral analysis methods are not well suited to describe highly nonsinusoidal signals, such as the skewed and asymmetric cycles that characterize time series of Middle and Late Pleistocene climate (Fig. 2) (Hagelberg et al., 1991; King, 1996; Lisiecki and Raymo, 2007). To mitigate the shortcomings of statistical tests that implicitly assume sinusoidality, higher-order spectral analysis techniques were introduced into the research field of palaeoceanography/-climatology in the 1990s (Hagelberg et al., 1991; Hagelberg et al., 1994; King Hagelberg and Cole, 1995; King, 1996).

Bispectral analysis was conceived in the 1960s within the research field that studies ocean waves (Hasselmann et al., 1963). It is an accepted method of quantifying nonlinear energy transfers among nearshore waves that reach breaking point when the seafloor shallows (Elgar and Guza, 1985; Doering and Bowen, 1995; Herbers et al., 2000; de Bakker et al., 2015). Since the pioneering interdisciplinary studies by palaeoceanographer/-climatologist Teresa (Terri) King Hagelberg and colleagues (notably the physical oceanographer/nearshore ocean wave researcher Steve Elgar) on Pleistocene and Holocene records (Hagelberg et al., 1991; Hagelberg et al., 1994; King Hagelberg and Cole, 1995; King, 1996), relatively few studies have applied higher-order spectral analysis methods to the palaeoclimate archive (e.g., (Muller and MacDonald, 1997b, a; von Dobeneck and Schmieder, 1999; Rial and Anaclerio, 2000; Rutherford and D'Hondt, 2000; Huybers and Wunsch, 2004; Huybers and Curry, 2006; Liebrand et al., 2017; Da Silva et al., 2018)). However, two more-recent developments make a re-appreciation for the potential of bispectra for palaeoclimate science timely. First, advancements have been made in the understanding and interpretation of the bispectrum (Herbers et al., 2000; de Bakker et al., 2015). These constitute (*i*) a shift of focus from bicoherence, which quantifies the strength of the couplings between the frequencies that are present in both the real and the imaginary parts of the complex-valued bispectrum, to (just) the imaginary part of the bispectrum, which can be used to compute nonconservative, relative energy exchanges, if time series are dominated by asymmetric wave forms/cycle shapes, and (*ii*) integration over the imaginary part of the bispectrum to quantify conservative, net energy transfers, and potentially absolute energy transfers, if net transfers can be scaled to the power spectrum (de Bakker et al., 2015; de Bakker et al., 2016). Second, increasingly noise-free benthic foraminiferal $\delta^{18}O$ stacks, characterized by precise and accurate age models, have become available to the palaeoclimate community (Lisiecki and Raymo, 2005; Ahn et al., 2017). Such data are a prerequisite for successful application of advanced higher-order spectral analysis methods, because they describe the distribution of nonsinusoidality (i.e., relatively small amounts of signal compared to the sinusoidal cycle properties) over time and frequency (King, 1996).

Both the 40-kyr problem of the Pliocene and Early Pleistocene, and the ~110-kyr problem (a.k.a. the 100-kyr problem) of the Middle and Late Pleistocene, are defined by considerable mismatches in spectral power, at their designated periodicities, between benthic foraminiferal $\delta^{18}O$ records (dominant) and summer insolation at any particular latitude (weaker or absent,

respectively) (Raymo and Nisancioglu, 2003; Raymo et al., 2006; Lisiecki, 2010). The mid-Pleistocene transition (MPT) (from ~1200 ka to ~600 ka) constitutes the temporal link between the 40-kyr and ~110-kyr "worlds", when climate cycles in benthic foraminiferal $\delta^{18}$O become longer in duration, of higher amplitude, and asymmetric in shape (Hagelberg et al., 1991; King, 1996; Lisiecki and Raymo, 2007). In association with this increasingly nonlinear response of Earth's climate-cryosphere system to radiative forcing, several climate cycles without a straightforward astronomical origin have been identified, such as those with periodicities of semi-precession, ~28-kyr, ~56-kyr, and ~80-kyr (Rutherford and D'Hondt, 2000; Berger et al., 2006; Lourens et al., 2010). Many, often contrasting, hypotheses have been proposed to address this complex climatic evolution of astronomically forced Pliocene and Pleistocene climate cycles, which range from the merging of the Cordilleran and the Laurentide ice sheets (Bintanja and van de Wal, 2008), a delayed isostatic rebound of the lithosphere-asthenosphere after land-ice mass loading (Abe-Ouchi et al., 2013), to regolith erosion (Clark and Pollard, 1998), and/or $P_{CO_2^{atm}}$ thresholds (Chalk et al., 2017), to name just a few (Roe and Allen, 1999; Huybers, 2011). However, more information needs to be extracted from geological archives to distinguish between these mechanisms. We examine the nonlinear response of the Pliocene–Pleistocene climate-cryosphere system to radiative forcing by applying state-of-the-art bispectral analysis techniques (Herbers et al., 2000; de Bakker et al., 2015; de Bakker et al., 2016) to one of the most noise-free palaeoclimate records, namely the LR04 benthic foraminiferal $\delta^{18}$O stack (Lisiecki and Raymo, 2005), with the aim to shed new light on (*i*) the nonlinear origins of climate cycles with and without a direct astronomical connection, (*ii*) the 40-kyr problem of the Pliocene and Early Pleistocene, (*iii*) the mid-Pleistocene transition, and (*iv*) the ~110-kyr problem of the Middle and Late Pleistocene. We show in great detail through which frequency interactions energy is transferred to the 40-kyr, the ~80-kyr and ultimately, ~110-kyr cycles. These new insights can be used to better link climate cycle geometries to nonlinear response mechanisms, and to obtain a better understanding of Earth's energy balance on astronomical time scales.

## 2. Methods

### 2.1. Benthic foraminiferal $\delta^{18}$O record of the Pliocene and Pleistocene

To quantify nonlinear energy transfers among astronomically paced cycles of Earth's climate-cryosphere system, we use the LR04 compilation of globally distributed records of stable oxygen isotope ratios ($\delta^{18}$O) measured on the calcite of benthic foraminifera (Lisiecki and Raymo, 2005). The $\delta^{18}$O values are given in per mille (‰) against the Vienna Peedee belemnite (VPDB). This stack spans the Pliocene–Pleistocene time interval, from 5,333 to 0 ka (Lisiecki and Raymo, 2005). We recompiled the original LR04 data set (39,473 data points, http://lorraine-lisiecki.com/stack.html) and resampled it at 1 kyr resolution using SiZer, a method that extracts structures in time series by assessing the statistically significant zero-crossing of its derivatives (Chaudhuri and Marron, 1999), because bispectral analysis requires a constant sampling resolution (Fig. 1

and Fig. 2). For the purposes of this study we selected the smooth (out of 41) that preserves most of the structure in the data, given the 1 kyr resampling resolution. This resampling resolution resulted in an oversampling of the original data for ~0.83% of the record (N = 328), most of which falls in the earliest Early Pliocene part of the record, which is excluded from the bispectral analysis for ages >5250 ka. The resultant resampled LR04 record is very similar in structure to the original LR04

stack (Lisiecki and Raymo, 2005), and, because of this similarity, we hereafter refer to the SiZer resampled LR04 record more simply as "LR04 stack". Prior to bispectral analysis we detrended the LR04 stack using a Gaussian Notch filter (frequency = 0.0 Myr$^{-1}$, bandwidth = 2.0 Myr$^{-1}$) to remove periodicities ≥500 kyr (Paillard et al., 1996). We also multiplied the age scale of the LR04 stack with −1, because the direction of time, which determines the sign of asymmetry values and the direction of energy transfers in bispectral analysis, increases toward the present. However, for the figures we follow the

convention, and plot ages that increase with geological age.

### 2.2. Quantifying geometries using central moments

Skewness and asymmetry are quantified using both the third central moment and bispectral methods (Fig. 2, Supp. Fig. 2, and Supp. Fig. 3) (see Section 2.3.3 for bispectral method) (Elgar, 1987). Excess kurtosis is quantified using the fourth

central moment only, because no trispectra were calculated (Fig. 2). Using third-moment quantities, skewness is determined by Eq. (1):

$$Sk(x) = \frac{\langle (x-\bar{x})^3 \rangle}{\langle (x-\bar{x})^2 \rangle^{3/2}},\qquad(1)$$

where the overbar indicates the mean value and where < > is the time averaging operator (i.e., window length over which the computation is performed) (Doering and Bowen, 1995), and asymmetry is determined following Eq. (2):

$$As(x) = \frac{\langle H(x-\bar{x})^3 \rangle}{\langle (x-\bar{x})^2 \rangle^{3/2}},\qquad(2)$$

where $H$ is the Hilbert transform (Kennedy et al., 2000). Skewness is a measure for the dissymmetry of a cycle relatively to a horizontal axis (Fig. 2b), whereas asymmetry is a measure for the dissymmetry of a cycle through time (Fig. 2c). Using fourth-moment quantities, excess kurtosis is defined by Eq. (3):

$$K(x) = \frac{\langle (x-\bar{x})^4 \rangle}{\langle (x-\bar{x})^2 \rangle^2} - 3.\qquad(3)$$

Excess kurtosis is a measure for the flatness or peakedness of the extrema of a cycle. Cycles with flat tops and bottoms (i.e., platykurtic cycles), such as a square-wave, have negative excess kurtosis values, whereas sharp-topped and sharp-bottomed cycles (i.e., leptokurtic cycles), such as a cardiogram, have positive kurtosis value (Fig. 2d).

## 2.3. Bispectral analysis

### 2.3.1. The bispectrum

In contrast to spectral analysis (e.g. using the fast Fourier transform), which gives the distribution of variance of a sinusoidal signal with frequency, bispectral analysis describes the distribution of nonsinusoidality with frequency (King, 1996). The
skewness of a cycle is described by the real part of the bispectrum, whereas the asymmetry of a cycle is deconvolved in the imaginary part of the bispectrum. The bispectrum shows nonconservative, relative energy exchanges among frequencies of a single time series. Energy transfers in the bispectrum computed on the LR04 stack are expressed in $‰^3$ $kyr^2$, which constitutes a statistical measure of energy. (We refer to Section 2.4.1. for an explanation of conservative, net energy exchanges, and to Section 4.2. for a discussion of the scaling of conservative to absolute energy exchanges). Energy transfers
can only occur, and be analysed by the bispectrum, if both the frequencies ($f$) and phases (Phi, $\varphi$, in radians) of these nonsinusoidal cycles are coupled: i.e., $f_1 + f_2 = f_3$ and $\varphi_1 + \varphi_2 = \varphi_3$, respectively (Hagelberg et al., 1991). Thus, for any possible combination of three frequencies the bispectrum assesses whether there is a coupling, and if so, whether energy exchanges occur. The transfer of energy in these so-called triad interactions is nonlinear, because the changes in cycle amplitudes ($A$) do not sum, and have a currently unknown, and probably variable, coupling through time (i.e., $A_1 + A_2 \neq A_3$,
see Section 4.2). We note that the meaning of "cycle", "frequency" and/or "periodicity" is different when referring to either the time or frequency domain. For example, a "single" cycle in a time series is often composed of multiple frequency components, which can be deconvolved by (bi)spectral analysis. However, for textual purposes, we use these words interchangeably, regardless of their reference to a specific domain.

The bispectrum is defined by Eq. (4):

$$B_{f_1, f_2} = E[C_{f_1} C_{f_2} C^*_{f_1 + f_2}], \tag{4}$$

where $E[…]$ is the ensemble average of the triple product of complex Fourier coefficients $C$ at the difference frequencies $f_1$, $f_2$, and their sum $f_3$ (i.e., $= f_1 + f_2$), and the asterisk indicates complex conjugation (Hasselmann et al., 1963). In this study, we focus on the imaginary part of the bispectrum (hereafter often referred to as "bispectrum", unless indicated otherwise),
following studies on ocean waves (e.g., (Herbers et al., 2000; de Bakker et al., 2015)), because according to bispectral theory most energy transfers are associated with asymmetric cycle shapes (i.e., wave forms), and because the climate cycles of the Middle and Late Pleistocene have high amplitudes (Fig. 1) and are highly asymmetric (Fig. 2). Furthermore, conservative, net energy exchanges between the coupled frequencies that are located in the imaginary part of the bispectrum can be computed if we assume a simple coupling coefficient between frequencies. Prior to bispectral analysis we detrend the data,
and subsequently apply a tapering function using a Hann (a.k.a. Hanning) window that also mitigates spectral leakage (Hagelberg et al., 1991), and then multiply the data with an energy correction factor to adjust for the change in amplitude that results from the windowing.

### 2.3.2. Interpreting the bispectrum

The $x$- and $y$-axes in the bispectrum, which correspond to $f_1$ and $f_2$ respectively, are mirror images of each other and share a symmetry axis at $x = y$ (Fig. 3) (Hasselmann et al., 1963). Therefore, only the positive, one-eighth part of the bispectrum is depicted, which is subdivided into 15 zones (see section 2.2.5, Table A1). For triad interactions only two outcomes exist: either, two frequencies (by definition: $f_1$ and $f_2$) gain energy (though not necessarily in equal measures), and one frequency (i.e., $f_3$) loses energy, or, $f_1$ and $f_2$ lose energy (ditto), and $f_3$ gains energy. Frequencies $f_1$ and $f_2$ are the so-called difference frequencies, while frequency $f_3$ is referred to as a sum frequency. Frequencies often participate in multiple interactions, and depending on the interaction, they act as either a difference frequency or a sum frequency. A particular frequency can thus receive energy through one interaction, and simultaneously lose energy in another interaction.

To help the interpretation of the bispectrum, we depict the main astronomical frequencies in the bispectrum with coloured lines (Fig. 3). Vertical and horizontal lines correspond to the difference frequencies ($f_1$ and $f_2$ respectively), and diagonal lines correspond to the sum frequency ($f_3$). Junctions between lines highlight the locations in the bispectrum where interactions occur between three periodicities (i.e., triad interactions) of which at least one is an astronomically-paced climate cycle, which can also interact with itself (Fig. 3, see also Table A2). No frequency axis is associated with sum frequency $f_3$, but its value can be read off at the crossing points with the $x$- and $y$-axes of the bispectrum, or by summing the $x$ and $y$ coordinates of the difference frequency axes at any point along the diagonals (Fig. 3).

We follow the convention, and define a triad interaction as negative or positive if $f_3$ either loses (blue colours) or gains (red colours) energy, respectively (Fig. 4). In written form, energy gains at a particular frequency are marked by an upward pointing arrow ($\uparrow$), and losses by a downward pointing arrow ($\downarrow$). Bispectral notation of triad interactions are given as $B_f(f_1, f_2, f_3)$ in the frequency domain (Myr$^{-1}$), and as $B_p(p_1, p_2, p_3)$ in the periodicity domain (kyr). For palaeoclimatological purposes we will mainly focus on the periodicities, which corresponding frequencies can easily be looked up in Table A2. We refer to the relevant part of the bispectrum with "$Re$" (i.e., real) and "$Im$" (i.e., imaginary), which are given in superscript ($B^{Re}$ and $B^{Im}$, respectively). Thus, if we consider a negative triad interaction, located in the imaginary part of the bispectrum, between $p_1$, a 40-kyr obliquity paced cycle (i.e., $f_1 = 25.0$ Myr$^{-1}$), and $p_2$, a ~95-kyr eccentricity paced cycle (i.e., $f_2 = 10.5$ Myr$^{-1}$), that results in a loss of energy at $p_3$, the ~28-kyr periodicity (i.e., $f_3 = 35.5$ Myr$^{-1}$), this is given as: $B_p^{Im}(40\uparrow, 95\uparrow, 28\downarrow)$ (i.e., $B_f^{Im}(25.0\uparrow, 10.5\uparrow, 35.5\downarrow)$) (Fig. 4a).

We note that within this study, we almost exclusively focus on the imaginary part of the bispectrum. This approach contrasts with previous palaeoclimate investigations that used the bicoherence spectrum (see e.g. (Hagelberg et al., 1991; von

Dobeneck and Schmieder, 1999; Wara et al., 2000; Huybers and Curry, 2006; Da Silva et al., 2018)). In short, bicoherence is a measure of the coherency between the real and imaginary parts of the bispectrum (i.e., where both parts have strong interactions), and is marked by positive values only (see e.g. (Elgar and Sebert, 1989; de Bakker et al., 2014)). Conversely, the real and imaginary parts of the bispectrum transform skewed and asymmetric cycle geometries into their composite
frequency components respectively, and both parts are characterized by positive as well as negative interactions.

### 2.3.3. Quantifying geometries using the bispectrum

Adopting the bispectral method to extract cycle geometries (Fig. 2, Supp. Fig. 2, and Supp. Fig. 3) (Elgar, 1987), skewness and asymmetry are computed from the biphase. The biphase reflects the ratio between the real and imaginary parts of the
bispectrum. The biphase is defined by Eq. (5):

$$Sk^{Re}(x) + As^{Im}(x) = \left[12 \sum_n \sum_l B(f_n, f_l) + 6 \sum_{p=1}^{\frac{N}{2}} B(f_p, f_p)\right] / E[x^2]^{3/2}, \qquad (5)$$

where $n$ and $l$ range from 1 to the Nyquist frequency N, with $n > l$ and $n + l \leq N$ (Elgar, 1987), and $f_p$ refers to the primary frequency (a.k.a. natural or resonance frequency, or first harmonic). We note that positive asymmetry values correspond to a dominance of negative interactions (i.e., a loss of energy at $f_3$) in the imaginary part of the bispectrum, and vice versa,
negative values for this geometry are indicative of an overall dominance of positive interactions (i.e., a gain of energy at $f_3$).

## 2.4. Power spectral density and net energy transfers

### 2.4.1. Integration of the spectrum and bispectrum

To quantify power spectral density, we apply a standard integration of the power spectrum. To obtain conservative, net
energy transfers per frequency (defined by the nonlinear source term $S_{nl}$), and hence, to extract the total gain or loss in energy per climate cycle over a specific time interval (i.e., window), we integrate over the imaginary part of the bispectrum and multiply it with a coupling coefficient. For palaeoclimate time series, no coupling coefficient has been determined previously. Therefore, we make minimum assumptions and use a coupling coefficient that only corrects for frequency $W_{(f_1, f_2)} = (f_1 + f_2)$. A comparable correction for frequency is also part of the Boussinesq scaling that is used for ocean
waves (e.g., (Herbers and Burton, 1997; Herbers et al., 2000)). The correction for frequency $(W_{(f_1, f_2)})$ insures energy conservation during triad interactions, because more energy is exchanged among the higher frequencies. Furthermore, this correction allows for qualitative interpretations of conservative net energy exchanges across the spectrum. If a more appropriate coupling coefficient for palaeoclimatic purposes were to be deduced, it could also scale the conservative net

energy transfers of the bispectrum (already corrected using $S_{nl}$ during integration) to absolute energy transfers that can be compared to (changes in) power spectral density (see Section 4.2). We express the integral of $S_{nl}$ in terms of an integration, or summation, over the positive quadrant of the bispectrum alone, or equivalently in sum and difference interactions following Eq. (6):

$S_{nl,f} = S_{nl,f^+} + S_{nl,f^-},$ (6)

where the sum contributions are expressed by Eq. (7):

$S_{nl^+} = \sum_{f \in F^+} W_{f',f-f'} \, I\{B_{f',f-f'}\},$ (7)

and the difference contributions are expressed by Eq (8):

$S_{nl^-} = -2\sum_{f \in F^-} W_{f+f',-f'} \, I\{B_{f,f'}\}.$ (8)

The sum contributions are obtained by integrating diagonally over the bispectrum, and the difference contributions are obtained by integrating along vertical and horizontal lines, perpendicular to the *x* and *y*-axes, respectively (Fig. 3). We track the evolution of net energy transfers through time by integrating over bispectra that are determined for consecutive and partially overlapping windows of the time series (i.e., a moving window) (Fig. 5, Fig. 6, and Supp. Fig. S3). Integration can be performed including all frequencies (i.e., over the entire imaginary part of the bispectrum), or over specific zones only, in

case subsets of interactions need to be further examined (Fig. 3, Table A1) (de Bakker et al., 2015; de Bakker et al., 2016). After integration over the bispectrum, either totally or zonally, energy transfers computed on the LR04 stack are expressed in ‰$^3$. Blue colours indicate a loss of energy at a specific frequency, red colours a gain.

### 2.4.2. Spectral and bispectral zones

To obtain more insight in the energy that is exchanged among eccentricity, obliquity and precession cycles we integrate the spectra and bispectra over separate zones (Fig. 3, Table A1). For bispectra, such a zonation approach was first applied in research concerned with nearshore ocean waves to distinguish between infragravity wave and sea-swell wave frequencies (de Bakker et al., 2015), and is adapted here to investigate the most important climate cycle bandwidths of the Pliocene and Pleistocene. The boundaries between the climate cycle zones are (arbitrarily) defined at frequencies of 17.8, 35.6 and 80.0

Myr$^{-1}$ (i.e., periodicities of ~56.2, ~28.1 and 12.5 kyr). In total, 15 bispectral zones are defined (Fig. 3, Table A1), each of which represents a unique combination of three main groups of astronomically-paced climate cycles and suborbital periodicities (i.e., supraorbital frequencies) that can interact and exchange energy with one other. We note that many periodicities without primary astronomical origin are included these "eccentricity", "obliquity", and "precession" zones. We focus here on the results of the eight most important zones, namely those involving astronomically-paced climate cycles

(Fig. 7, Fig. 8, Fig. 9, Supp. Fig. S4 and Supp. Fig. S5). To better highlight the roles of all nonlinear interactions involving eccentricity, obliquity, or precession, we recombine (i.e., sum) the individual zones (Fig. 10).

## 3. Results

### 3.1. Geometries of Pliocene and Pleistocene climate cycles

The geometry computations confirm the qualitative visual inspection of the LR04, globally averaged, benthic foraminiferal $\delta^{18}O$ stack, and show that strong nonsinusoidal cycle shapes are present in the data, especially during the Middle and Late Pleistocene part of the record (Fig. 2) (Lisiecki and Raymo, 2007). We focus the geometry interpretations on the time interval with the greater amplitude variability in the $\delta^{18}O$ record, which start to increase (gradually) from ~3,000 ka onward. Between ~3,000 and ~350 ka, the skewness of climate cycles varies between −0.5 and 1.0. Peak values of ~1.0 are reached between ~2,500 and ~1,500 ka, after which skewness rapidly decreases at ~1,500 ka to negative values of −0.5 that slowly increase to 0.0 during the MPT (Fig. 2). Glacial-interglacial cycle asymmetry varies between about −0.5 and 1.5. The most prominent steps in asymmetry occur at ~2,500 and ~1,000 ka (the latter during the MPT), when values increase from about −0.5 to 0.5 and from approximately −0.2 to 1.2, respectively. These results compare well to those previously obtained on (stacks of) benthic foraminiferal $\delta^{18}O$ records (Hagelberg et al., 1991; King, 1996; Lisiecki and Raymo, 2007). Excess kurtosis computations identify leptokurtic (thin-peaked) cycle shapes that vary between 0.5 and 3.0 from ~3,000 ka onward. Unlike skewness and asymmetry, which are marked by multi-Myr decreasing and increasing trends, respectively, no long-term trend can be discerned in Pleistocene cycle kurtosis (Fig. 2).

### 3.2. Bispectra of Pliocene and Pleistocene climate cycles

Nonsinusoidal cycle shapes may indicate the potential for obtaining more information about nonlinear interactions among frequencies using higher order spectral analysis (Hagelberg et al., 1991; King, 1996). We present three examples of Pliocene and Pleistocene bispectra (i.e., their imaginary parts), based on the LR04 stack, which are characterized by clear triad interactions (Fig. 4, see also Section 2.2.2.). The first example is a bispectrum across the Middle to Late Pleistocene (i.e., the "~110-kyr world"), which shows clear and punctuated triad interactions concentrated where $p_2$ equals 95 kyr (Fig. 4a). The main interactions are located at the following triads, or their connecting lines: at ~$B_p^{Im}$(95↑, 405↑, 77↓), weakly at $B_p^{Im}$(95↑, 125↑, 54↓), from $B_p^{Im}$(95↑, 95↑, 48↓) via $B_p^{Im}$(69↑, 95↑, 40↓) to $B_p^{Im}$(80↑, 80↑, 40↓), broadly around $B_p^{Im}$(40↑, 95↑, 28↓), at $B_p^{Im}$(40↑, 51↑, 22↓), from $B_p^{Im}$(31↑, 95↑, 24↓) to $B_p^{Im}$(29↑, 95↑, 22↓), and weakly (but still visible) at the triple junction $B_p^{Im}$(24↑, 95↑, 19↓). Interactions at these triads (Table A2) are all negative, indicating that energy is mainly transferred to the 95 kyr (and 125 kyr) cycle from the three precession periodicities, the ~28-kyr periodicity and the 40-kyr obliquity periodicity. A very weak positive interaction is located at $B_p^{Im}$(125↓, 405↓, 95↑). Strong energy gains at the ~110-kyr periodicity of eccentricity modulated precession thus characterize the bispectrum of the Middle to Late Pleistocene interval.

The second example is a bispectrum across the mid-Pleistocene transition (i.e., the "~80-kyr world"), and shows several negative interactions where either $p_1$ or $p_2$ is equal to ~80 kyr (Fig. 4b). Particularly strong negative interactions are located at ~$B_p^{Im}(80\uparrow, 200\uparrow, 57\downarrow)$, $B_p^{Im}(80\uparrow, 80\uparrow, 40\downarrow)$, and at $B_p^{Im}(40\uparrow, 80\uparrow, 27\downarrow)$. Also $B_p^{Im}(40\uparrow, 58\uparrow, 24\downarrow)$ is strongly negative. We document one major positive interaction at $B_p^{Im}(69\downarrow, 95\downarrow, 40\uparrow)$, which connects to $B_p^{Im}(59\downarrow, 125\downarrow, 40\uparrow)$. Due to the relatively loosely distributed triad interactions in this bispectrum (Fig. 4b), compared to, for example, the bispectrum of the Late Pliocene to Early Pleistocene (Fig. 4c), it is more difficult to interpret dominant energy gains and losses directly from the bispectrum of the MPT. However, we observe an overall gain of energy at the 70- to 80-kyr periodicities, and a dual role for the 40-kyr periodicity, which operates as both energy receiver and supplier simultaneously. We note that the ~80-kyr periodicity is not an obvious triad of the primary astronomical cycles (Table A2), apart from the junction at $B_p^{Im}(95\uparrow, 405\uparrow, 77\downarrow)$, where no strong interactions are documented in this particular bispectrum. The 80-kyr periodicity thus starts as a "lower" harmonic or "subharmonic" (i.e., where lower and sub refer to the frequency domain) of the 40-kyr periodicity, which in turn is a first harmonic of obliquity forcing, as well as a "combination tone" of precession and nonastronomical periodicities (Table A2) (Pestiaux et al., 1988). This subharmonic origin for the 80-kyr periodicity is supported by the strong negative interaction at $B_p^{Im}(80\uparrow, 80\uparrow, 40\downarrow)$.

The third example is a bispectrum across the Late Pliocene to Early Pleistocene interval (i.e., the "40-kyr world"), which depicts negative and positive interactions, where both difference frequencies $f_1$ and $f_2$, and sum frequency $f_3$ are equal to 25 Myr$^{-1}$ (i.e., 40 kyr), respectively (Fig. 4c). More specifically, negative interactions are concentrated at and between triads along the lines from $B_p^{Im}(40\uparrow, \infty\uparrow, 40\downarrow)$ to $B_p^{Im}(40\uparrow, 40\uparrow, 20\downarrow)$, and from $B_p^{Im}(40\uparrow, 40\uparrow, 20\downarrow)$ to $B_p^{Im}(36\uparrow, 40\uparrow, 19\downarrow)$, indicating a gain at the 40-kyr periodicity. Most notable is the strong negative interaction at ~$B_p^{Im}(40\uparrow, 93\uparrow, 28\downarrow)$ to ~$B_p^{Im}(40\uparrow, 83\uparrow, 27\downarrow)$. Other bispectral coordinates marked by negative interactions are those between $B_p^{Im}(95\uparrow, 95\uparrow, 48\downarrow)$ and ~$B_p^{Im}(70\uparrow, 150\uparrow, 48\downarrow)$. Positive interactions are concentrated on the line between the bispectral coordinates $B_p^{Im}(80\downarrow, 80\downarrow, 40\uparrow)$ and $B_p^{Im}(44\downarrow, 405\downarrow, 40\uparrow)$, which also indicate a gain at the 40 kyr periodicity by transfers from numerous other frequencies. Overall, the Late Pliocene to Early Pleistocene bispectrum depicts strong energy gains at the obliquity paced 40-kyr periodicity through many triad interactions, some of which are with other astronomically-paced climate cycles.

### 3.3. Qualitative comparison of summer-half insolation to the climate-cryosphere proxy record

Figure 5 shows three time-evolutive spectral and bispectral analyses for the Pliocene-Pleistocene time interval. It compares the total power spectral density of summer-half insolation at 65°N (Fig. 5a) (Laskar et al., 2004) to the total energy transfers and total power spectral density of the LR04 benthic foraminiferal $\delta^{18}O$ stack (Fig. 5b and Fig. 5c, respectively). This insolation curve constitutes a classical shorthand for the complex external forcing of the ice ages (not shown, but see Fig. 1e and Fig. 2e for individual records of precession, obliquity and eccentricity) (Köppen and Wegener, 1924; Milankovitch, 1941). Figure 6 highlights five, equally spaced (500 kyr apart), time-slices of the three panels of Figure 5. Figure 7 depicts three, zonally averaged frequency-slices of the three panels of Figure 5; namely the "eccentricity", "obliquity", and "precession" zones (see Section 2.4.2.), as well as the total of all frequencies combined.

### 3.3.1. Power spectral density of insolation

The time-evolutive power spectra of insolation, are marked by high spectral power at the precession and obliquity periodicities, and the near-absence of spectral power at the eccentricity periodicities (Fig. 5a, Fig. 6, and Fig. 7a) (Hays et al., 1976; Berger, 1977). Furthermore, it is characterized by amplitude variability in spectral power at these periodicities, which is governed by the long term (eccentricity) modulations of obliquity and precession. These amplitude modulations have durations of ~180 kyr (not detected here, due to window length) and 1.2, 2.4 and 4.8 Myr for obliquity, and 405 kyr and 2.4 Myr for eccentricity modulated precession (Fig. 5a and Fig. 7a). Concurrent with the MPT we document the sharpest decrease in total spectral power of the entire Pliocene and Pleistocene, which is largely due to a decrease in precession power from maximum values (Fig. 7a) (Laskar et al., 2004). This is indicative of a reduction in the perturbation of the Earth's system that results from a smaller difference in insolation forcing between the hemispheres (i.e., reduced seasonality).

### 3.3.2. Energy transfers among climate cycles

Figures 5b and S3 depict the conservative net energy transfers among (astronomically-forced) climate cycles as present in the LR04 stack. The net energy transfers show how energy is redistributed over the power spectrum among astronomically-paced and nonastronomically-paced climate cycles. They reveal strong increases in both energy gains (in red) and losses (in blue) toward the present, and from ~1,000 ka onward several consecutive shifts in energy transfers from lower to higher periodicities occur (Fig. 5b). Energy gains are highly concentrated at the 40-kyr periodicity from ~2,500 ka onward and are marked by a modest 1.2 Myr beat in amplitude (Fig. 5b and Fig. 7b). A second location of strong energy gains is the ~80-kyr periodicity from ~1,000 ka onward (Fig. 6b), which at ~700 ka shifts (Fig. 5b) to a dominant gain for the 95-kyr component

(Fig. 6a). Further energy gains are located at a ~200-kyr periodicity from ~750 ka onward, a 50-kyr to 60-kyr periodicity between ~1500 and 500 ka, and (very weak) at the 24-kyr periodicity from ~650 ka onward (Fig. 5b). Energy losses are also observed in the total integration over the bispectra (Fig. 5b, Fig. 6 and Fig. 7b). However, these losses are generally of smaller amplitude per cycle and are distributed across a greater number of periodicities, in comparison to the highly concentrated energy gains (Fig. 5b and Fig. 6). Despite this more dispersed pattern of energy losses, we highlight the more conspicuous decreases: the 40-kyr periodicity from ~550 ka onward, the 30- to 36-kyr periodicities between 1,500 and 1,000 ka, the 30- to 26-kyr periodicities between ~2,500 and ~2,000 ka, the ~28-kyr periodicity from ~800 ka onward, the 24-kyr periodicity between ~1,000 and ~800 ka, the 22-kyr periodicity from ~700 onward, the 19-kyr periodicity from 2,000 ka onward, and the 15-kyr periodicity intermittently from ~1,000 ka onward (Fig. 5b).

### 3.3.3. Power spectral density of climate cycles

Time-evolutive power spectral analysis of the LR04 record is characterized by high amplitude variability at the 40-kyr obliquity-paced cycle, from approximately 4,000 ka onward (Fig. 5c, Fig. 6, and Fig. 7c). In addition, distinct ~80 and ~110 kyr cycles of rapidly increasing amplitude are documented from ~1,200 ka onward, marking the MPT (Fig. 6b) and post-MPT (Fig. 6a) intervals, respectively (Lisiecki and Raymo, 2005, 2007). We note that the 125 kyr component of the ~110 kyr signal in the power spectra (Fig. 5c), is not present in the integration over the bispectrum (Fig. 5b) A relatively weak ~200-kyr cycle is present from ~750 ka onward (Fig. 5c). No strong precession response is present (Raymo and Nisancioglu, 2003; Raymo et al., 2006). The 40-kyr cycle in $\delta^{18}O$ partially mimics the 1.2 Myr amplitude modulation of the obliquity cycle, especially from ~3,000 ka onward (Fig. 5c) (Lourens and Hilgen, 1997; Lisiecki and Raymo, 2007). We draw attention to a subtle "arc" in spectral power (absent in bispectral power), which starts with a strong 405-kyr cycle, at ~3.500 ka, that becomes gradually shorter in duration until it resonates with the 95-kyr eccentricity modulation of precession, at ~2,650 ka, and then culminates in several weaker 70 to 80 kyr cycles at ~2,000 ka (Fig. 5c, see also Fig. 1b) (Lisiecki, 2010; Meyers and Hinnov, 2010; Viaggi, 2018).

### 3.4. Zooming in on energy transfers

To better understand the different roles of specific (bandwidths of) climate cycles in nonlinear interactions, we determine conservative net energy transfers per bispectral zone (Fig. 8, Fig. S4, and Fig. S5). These zones reflect the "layers" that make up total energy exchanges as depicted in Figure 5b. We assess if triad interactions are conservative of energy for each of the bispectral zones and for the entire bispectrum (Fig. 9). Lastly, we recombine the bispectral zones again for frequency

bandwidths involving precession-, obliquity- or eccentricity-paced climate cycles to highlight their influences in nonlinear interactions (Fig. 10).

### 3.4.1. Zonal energy transfers

In Zone 1, from 1,000 ka onward, we document a modest exchange of energy between the (eccentricity related) periodicities in the range from ~200 to 50 kyr (Fig. 8a). Notably, from ~500 ka onward, periodicities of ~80 to 50 kyr predominantly lose power, whereas those between ~200 and ~80 kyr mainly gain energy. Stronger exchanges are documented in Zone 2 (Fig. 8b). Especially the 40-kyr obliquity paced cycle gains energy between 3,500 and 900 ka in this zone, after which it loses energy from 900 ka onward. Also, the 50 to 60-kyr periodicity loses energy, especially from ~600 ka onward. Between ~1,100 and ~800 ka a single periodicity of ~87 kyr gains power. At ~750 ka this periodicity splits into an ~80-kyr periodicity and a 95-kyr eccentricity-paced cycle, which both gain energy. In general, interactions in Zone 2 change from (weakly) positive ($B^{im}$(E↓, E↓, O↑)) to negative ($B^{im}$(E↑, E↑, O↓)) during the MPT (~1,000 ka), indicative of a change from escalating energy to higher frequencies, to a cascading from higher frequencies. We note that energy exchanged through $B^{im}$(E, E, O)-interactions (Zone 2) are almost of equal amplitude as those of all other interactions combined. Zones 3 to 5 are mainly marked by increases in energy at the 40-kyr cycle, roughly from 2,500 ka onward (Fig. 8c to Fig. 8e). A smaller gain is documented in these zones (especially in Zone 3) at periodicities between ~70 and 50 kyr from ~1,300 ka onward, and in Zone 3 at the 95 kyr cycle from ~650 ka onward. Furthermore, we document decreases in energy at the ~30 to 27-kyr periodicities, the ~28 to 22-kyr periodicities, and the main precession periodicities of 24, 22 and 19-kyr, for Zones 3, 4, and 5, respectively. The switchover during the MPT, at about 750 ka, of energy loss at the 24-kyr cycle to a loss at the ~28-kyr periodicity is associated with $B^{Im}$(O, E, P)-interactions (Zone 4) only. Zones 6 and 7 show weaker energy exchanges among the main astronomical periodicities. In Zone 6 the ~27 and 24-kyr cycles show a small gain between ~1,000 and 700 ka, and from ~700 ka onward, respectively. We only document very minimal direct fuelling of eccentricity-paced climate cycles by precession-paced climate cycles in this zone. Energy is lost in Zone 6 from 1,500 ka onward at the 19-kyr cycle. In Zone 7 relatively weak gains are present at the main obliquity and precession cycles from 1,500 ka onward. Energy losses of Zone 7 are located mainly at sub-precession periodicities of ~15 to <13 kyr (Fig. 8f and Fig. 8g). No energy exchanges of similar magnitude are documented in Zone 8 (Fig. 8h).

Overall, the zonal integrations are marked by negative triad interactions, and thus reveal a cascade of energy from the (sub-) precession bandwidth, to the obliquity and (ultimately) eccentricity bandwidths through several successive interactions (Fig. 8). These analyses further show that many periodicities fulfil a dual role that often remains stable through time, namely: they serve simultaneously and persistently as energy provider and receiver. For example, the 24-kyr precession-paced cycle gains energy through $B^{Im}$(P, E, P)- and $B^{Im}$(P, O, P)-interactions, as documented in Zones 6 and 7, meanwhile losing energy

through $B^{Im}$(O, O, P)- and $B^{Im}$(O, E, P)-interactions, as can be observed in Zones 4 and 5. As an exception to this persistent behaviour of many lower periodicities in transferring energy to higher periodicities, the role of the 40-kyr obliquity-paced cycle evolves, but only in $B^{Im}$(E, E, O)-interactions (Zone 2). In Zones 3, 4, 5, and 7 the 40-kyr obliquity-paced climate cycle (predominantly) partakes in triad interactions as a difference frequency, and it continuously receives energy, mainly from precession. However, during the MPT (from ~900 ka onward) in Zone 2, when the 40-kyr periodicity partakes in interactions solely as a sum frequency ($f_3$) of two eccentricity components ($f_1 + f_2$), it starts providing energy to the eccentricity bandwidth. The net (i.e., total) result of these opposing roles for the 40-kyr obliquity paced cycle after the MPT is that energy losses overtake the gains (Fig. 5b and Fig. S3). However, the initial decrease of the integrated "obliquity" zone, between ~1,000 and 600 ka, is linked to the ~28-kyr periodicity, not the 40-kyr periodicity (Fig. 5b).

### 3.4.2. Energy conservation

Computations of energy conservation, hereafter also referred to as "conservativity", indicate that both the total as well as the zonal energy gains in triad interactions are commensurate with losses, given our assumed coupling coefficient (Fig. 9). The dominance of negative interactions (i.e., positive values for $f_1 + f_2$, and negative values for $f_3$) is indicative of a transfer within the Earth's climate-cryosphere system of insolation energy to lower frequencies. Energy is not well conserved in triad interactions documented by Zone 8. This is probably the result of the low number of interactions located in Zone 8 (Fig. 4), and the small area of this zone (Fig. 3).

### 3.4.3. Energy exchanges involving precession-, obliquity-, or eccentricity-paced climate cycles

To highlight the separate roles of the precession, obliquity and eccentricity bandwidths during nonlinear interactions within the climate-cryosphere system, we recombine (i.e., sum) their respective bispectral zones of Figure 8. To do so, we include those zones that contain either one, two or three frequency-components of a particular bandwidth (Fig. 3 and Fig. 10). The general pattern of an energy cascade is robust, but it becomes clearer that precession cycles do not contribute much to the fuelling of eccentricity-paced climate variability directly (Fig. 10a–c). The 40-kyr periodicity during the Middle and Late Pleistocene gains energy from precession dominated interactions, but loses energy in obliquity and eccentricity dominated interactions (Fig. 10a–c). A comparison of conservativity of the recombined zones that include at least one precession, obliquity, or eccentricity component, indicates that approximately similar amounts of energy are exchanged in (triad) interactions involving obliquity, as in those involving eccentricity (Fig. 10d).

## 4. Suitability of bispectral applications for palaeoclimatology

### 4.1. Limitations

#### 4.1.1. Age uncertainty and signal to noise ratio

By combining more than fifty individual benthic foraminiferal $\delta^{18}O$ records, the LR04 stack simultaneously decreases the age uncertainty and increases the ratio of signal-to-noise considerably in comparison to any record individually (Lisiecki and Raymo, 2005). Reduced age uncertainties and improved signal-to-noise ratios of stacked records may be obtained through the computation of averaged sedimentation rates in combination with statistical "binning" (Lisiecki and Raymo, 2005), a probabilistic approach using a hidden Markov model (Ahn et al., 2017), or, as is the case in this reanalysis of the (1-kyr resampled/binned) LR04 stack, by additional statistical "smoothing" (Chaudhuri and Marron, 1999). For successful application of advanced bispectral analyses techniques, good age control and (very) high signal to noise ratios are needed, because the distributions of only small amounts of signal over time and frequency are considered (Hasselmann et al., 1963; King, 1996). For consecutive computational steps that use bispectra (i.e., the bispectrum and the integration over the bispectrum), increasingly better constrained and noise-freer data are required, because each step zooms in further on these small amounts of signal that are associated with nonsinusoidal geometries. The bispectra of climate cycles presented here show clear interactions (Fig. 4), and the integrations of the bispectrum reveal qualitatively consistent patterns of energy transfers for analysis with varying window lengths (Fig. 5b and Supp. Fig. 3). Therefore, we argue that the LR04 stack has sufficiently accurate age calibrations, and a high enough signal-to-noise ratio for the successful application of advanced bispectral analysis.

#### 4.1.2. Resolvability in time and frequency domains

Similar to the power spectrum, the bispectrum is marked by a trade-off between resolution in the time versus frequency domains (Fig. S3). Bispectral results become more significant, and can yield greater degrees of freedom, if a larger number of similarly shaped cycles, or waveforms, can be included for analysis (i.e., a lengthening of the window). For example, studies on natural nearshore waves or those on flume-generated waves (periodicities of seconds to minutes), use hour-long stable time series with high sampling resolutions and wave numbers (de Bakker et al., 2014). Bispectral analyses of such datasets are well resolved in the frequency domain. Furthermore, they permit the computation of robust confidence levels on the results, and the selection of statistical settings that yield high degrees of freedom (e.g., (Elgar and Guza, 1985; de Bakker et al., 2015)). However, for the Pliocene and Pleistocene record, similar statistical objectives are unattainable, because no two climate cycles are alike and because of the low number of ~110-kyr cycles in the Pleistocene (King, 1996; Lisiecki,

2010). As a result of this absence of "ergodicity" (i.e., a stable waveform through space and time), which characterises the unique Pliocene-Pleistocene climatic evolution as captured by the LR04 stack, relatively short window-lengths are preferred to obtain well-resolved results in the time domain. The Hann windowing function visually clarifies the interactions in both the bispectra as well as the integrations over the bispectrum. Similarly to previous bispectral studies on the palaeoclimate archive (Hagelberg et al., 1991; King, 1996), we cannot define confidence levels for the bispectral analysis of the LR04 stack, because the number of cycles per window is too small. However, the bispectral couplings among climate cycles documented here are robust for varying bispectral settings (Fig. 5, Fig. 8, and Supp. Fig. S3 to Supp. Fig. S5).

### 4.1.3. Quantifying cycle geometries of nonergodic records

In contrast to the relatively more robust bispectral results (see previous section), absolute geometry values are more strongly affected by the nonergodic nature of the Pliocene-Pleistocene record. Choices of (*i*) window length, (*ii*) method of detrending the time series, and (*iii*) the application of a windowing function or not, can have profound effects on the robustness of the results (van Peer, 2018). These somewhat arbitrary choices affect both ways of quantifying geometries similarly (i.e., using the central moments or higher order spectra (Fig. 2, Supp. Fig. 2, and Supp. Fig. S3) (van Peer, 2018). The sensitivity of geometry computations to data processing techniques was not previously appreciated in full, and may have resulted in over-optimistic confidence levels for cycle geometry values computed on an Oligocene and early Miocene climate record (Liebrand et al., 2017). We note that the uncertainty in skewness and asymmetry becomes larger for cycles with values near the mean (= 0). The relatively high values of about –1 or +1, similar to those computed on the LR04 stack of the Middle and Late Pleistocene, are much less sensitive to data processing methods, and have been independently computed on at least three occasions (Fig. 2) (Hagelberg et al., 1991; King, 1996; Lisiecki and Raymo, 2007). Remarkably, excess kurtosis values between ~3,000 and ~350 ka even change sign between computations that do apply a Hann window (resulting in leptokurtic cycles; Fig. 2) and those that do not (yielding platykurtic cycles that reach values of −1.0 during the Pleistocene; not shown). By comparison, skewness and asymmetry values appear much more robust to various methods of data processing than values of excess kurtosis. This greater sensitivity of excess kurtosis to windowing is not surprising considering that fourth moment statistics (and the trispectrum) zoom in on even smaller amounts of signal in comparison to the third central moments and the bispectrum (Collis et al., 1998).

### 4.2. Coupling coefficient

For ocean waves, the relationship between the integration of the bispectrum and absolute nonlinear energy transfers is determined by a coupling coefficient, which is based on the second-order Boussinesq theory (Herbers and Burton, 1997;

Herbers et al., 2000). This Boussinesq approximation describes how energy is transferred from the primary ocean waves to higher and lower frequency components by near-resonant nonlinear triad interactions. A coupling coefficient is needed (*i*) to ensure the conservation of energy within a single triad interaction (i.e., all energy lost at a certain frequency is compensated by an energy gain at the other two frequencies, and vice versa), and (*ii*) to obtain absolute energy transfers, which are directly comparable with the changes in the power spectrum when the waves propagate toward the coast (Herbers et al., 2000; de Bakker et al., 2014). Here, we apply a similar approach to palaeoclimate time series, and demonstrate that energy conservation can be ensured assuming a simple coupling coefficient that multiplies the energy transfers at each specific sum frequency after integration, with that specific frequency ($f_3$) (see Section 2.4.1). Comparison of energy gains and losses, for both the integration over the entire imaginary part of the bispectrum, as well as for each bispectral zone separately, shows that sum ($S_{nl+}$, see Eq. 7) and difference ($S_{nl-}$, see Eq. 8) interactions balance each other, which is indicative of energy conservation, and qualitatively trustworthy signals in energy transfers (Fig. 9). However, the second step toward absolute energy transfers is not developed here (see qualitative comparison of spectral and bispectral results in Fig. 6), because a more fundamental understanding is needed of the many climatologic and oceanographic, biologic, sedimentologic and lithologic processes that affect the (globally integrated) benthic foraminiferal $\delta^{18}O$ record. Ultimately, these poorly constrained processes underpinning the LR04 stack determine how to scale bispectral energy transfers (in $‰^3$) to changes in power spectral density (in $‰^2$ kyr) in absolute terms.

## 5. Discussion

### 5.1. Revisiting the problem of the ice ages

The common denominator of (*i*) the 40-kyr problem of the Pliocene and Early Pleistocene, (*ii*) the ~110-kyr problem of the Middle and Late Pleistocene, and (*iii*) the (nonlinear) origins of many climate cycles without primary astronomical pacemaker (e.g., the ~28-kyr and 80-kyr periodicities), is the mismatch in the distribution of power spectral density between benthic foraminiferal $\delta^{18}O$ and records and insolation (Raymo and Nisancioglu, 2003; Lisiecki, 2010; Lourens et al., 2010). Furthermore, (*iv*) during the mid-Pleistocene transition, this discrepancy increases significantly (Lisiecki and Raymo, 2005; Clark et al., 2006; Huybers, 2007), by the shifting of spectral power of climate records from the 40-kyr, via the ~56-kyr and the ~80-kyr, to the ~110-kyr periodicity (Rial and Anaclerio, 2000; Lourens et al., 2010; Viaggi, 2018). In light of the bispectral results obtained in this study, we revisit the problem of the ice ages (Agassiz, 1840; Milankovitch, 1941; Hays et al., 1976), in search for the most parsimonious explanation(s) for the origins and evolution of climate cycles with and without astronomical periodicities. The answer that we distil from the bispectra of Pliocene and Pleistocene climate cycles, in greater clarity than before (Hagelberg et al., 1991; Rial and Anaclerio, 2000; Lourens et al., 2010), is that these problems (i.e., *i* to *iv*) are intrinsically linked by the (many) increasingly nonlinear responses of Earth's climate-cryosphere system to

insolation forcing (Pisias et al., 1990; Yiou et al., 1994). Namely, the bispectra confirm (Hays et al., 1976; Hagelberg et al., 1991), that energy is transferred from the lower to the higher periodicities from at least 2,500 ka onward. More importantly, they show (*i*) how this happens, i.e., through which nonlinear triad interactions (Fig. 4), and (*ii*) how their involvement in transferring energy evolves trough time (Fig. 5, Fig. 8, and Fig. 10), especially during MPT.

## 5.2. The origins of climate cycles: toward a better mechanistic understanding

To advance the understanding of the origins of Pliocene and Pleistocene climate cycles, we explore the potential climatic and cryospheric processes that cause the documented nonlinear triad interactions and energy transfers. Similar to the difficulties in linking spectral peaks of benthic foraminiferal $\delta^{18}O$ records, which describe the integrated effect of deep water

temperatures and cryosphere (Hays et al., 1976; Elderfield et al., 2012; Rohling et al., 2014; Crowhurst et al., 2018), to particular mechanisms, it is not straightforward to identify (singular) causes for bispectral peaks (Fig. 4). Should we attempt to link every individual triad interaction to a specific climatic or cryospheric process? Or, alternatively, should we search for one, or a perhaps a couple of, mechanism(s) that can explain multiple, or even all, nonlinear couplings at once? A comparison to nearshore ocean waves, and the interactions that characterize them as they break in the surf zone, may be

informative (Herbers et al., 2000; de Bakker et al., 2014, 2016). The strength and number of nonlinear interactions of shoaling and breaking waves is strongly determined by the shallowing of the water depth when the wave approaches the coast, and starts to interact with the sea bed (de Bakker et al., 2015; de Bakker et al., 2016). Thus, a single mechanism, i.e., the interaction between a wave and the beach, causes multiple nonlinear triad interactions.

In contrast to the single cause for multiple nonlinear triad interactions among ocean waves, we tentatively link two mechanisms to energy transfers among climate cycles, because we observe two key features in the total and zonal integrations over the bispectrum: (*i*) a persistent fuelling of obliquity-paced climate cycles by precession-paced climate cycles, which we link to astronomical forcing of atmospheric and oceanic circulation ("the climatic precession motor") (Section 5.2.1), and (*ii*) a change of obliquity-paced climate cycles from net energy sink into net energy source, which we

link to a resonance of cycles of crustal sinking and rebounds with the eccentricity modulation of precession, after NH land-ice mass-loading passed a critical threshold during the MPT ("the cryospheric obliquity motor") (Section 5.2.2.).

## 5.2.1. The climatic precession motor

First, we speculate that the fuelling of the ice ages by (mainly) the precession periodicities is largely linked to the low-to-mid

latitude monsoons (Rutherford and D'Hondt, 2000; Berger et al., 2006; Bosmans et al., 2015a). The (sub-) tropical zones,

where most insolation is received, constitute Earth's heat engine, and their monsoons function as both a source of moisture and as a meridional teleconnection to build up large polar (land-) ice volumes. The interactions between NH land-ice volumes and the African and Asian monsoons are well documented for large parts of the Pliocene and Pleistocene (deMenocal, 1995; Sun et al., 2006; Cheng et al., 2016). The monsoonal (i.e., atmospheric) redistribution of heat and
moisture was most probably aided by orbital scale variability in the strength of Atlantic (i.e., oceanic) meridional overturning circulation (AMOC) (Lisiecki et al., 2008; Ziegler et al., 2010; Elderfield et al., 2012). Based on the bispectral results, we infer that during the Pliocene and Early Pleistocene this predominantly monsoonally-driven precession motor fuels the 40-kyr obliquity-paced ice age cycles, aided by more linear climatic-cryospheric responses resulting from variability in insolation at this periodicity, either zonally/cross-equatorially integrated (Huybers and Tziperman, 2008; Bosmans et al.,
2015b), or locally at higher latitudes. The precession motor is especially strongly expressed in $B^{Im}$(O, O, P)-interactions (Zone 5, Fig. 8e). However, uncertainty remains over the exact moisture sources for the Pleistocene land-ice volume fluctuations (e.g. (Werner et al., 2001; de Boer et al., 2012)).

### 5.2.2. The cryospheric obliquity motor

Second, we hypothesize that the role reversal of obliquity paced climate cycles (from net energy receiver to net energy provider) across the MPT, in both $B^{Im}$(E, E, O)-interactions (Fig. 4) and total net energy transfers (Fig. 5b and Fig 8b), is linked to a resonance of (auto-) cycles of crustal sinking and a delayed rebound with ~110-kyr long (allo-) cycles of eccentricity modulated precession. In this hypothesis, crustal sinking is triggered by the passing of a land-ice mass-loading threshold, and the delayed response to both eccentricity modulated precession and obliquity, is caused by the inertia of
bulky, continental-sized ice sheets, which also results in a nonlinear positive ice albedo-feedback, and hence, a hysteresis-pathway for glacial-interglacial cycles (Calov and Ganopolski, 2005; Abe-Ouchi et al., 2013). This obliquity motor evolves during and after the MPT, from a lower harmonic of two to three obliquity-paced cycles (Fig. 6b) (Ridgwell et al., 1999; Huybers and Wunsch, 2005), into a resonance (i.e., phase coupling/locking, or synchronization) with the ~110-kyr eccentricity paced cycles, mainly the ~95-kyr component (Fig. 6a) (Hagelberg et al., 1991; Tziperman et al., 2006; Crucifix,
2012; Abe-Ouchi et al., 2013). We speculate that the resonance of crustal sinking and a delayed isostatic rebound of the lithosphere-asthenosphere, also shifted the sensitivity of the climate-cryosphere system to insolation forcing at higher latitudes, where the relative influence of obliquity compared to precession is greater than at lower latitudes. We conjecture that this shift in sensitivity to obliquity-forcing was especially enhanced during glacial terminations (Huybers and Wunsch, 2005; Huybers, 2011; Konijnendijk et al., 2015).

## 5.3. Climatic and tectonic boundary conditions

If our interpretations of the bispectra are indeed supportive of an astronomically-paced monsoonal/AMOC "push" of (heat and) moisture, followed by a resonant lithospheric-asthenospheric "pull" through a cooling of the NH polar region after an initial phase of ice growth, for sufficiently large glacial land-ice masses to form, then they would highlight the importance of delayed isostatic rebound and the shape (i.e., extent, volume, and especially height) of the NH ice sheets in determining the pattern and geometry of climate-cryosphere variability observed for the Middle and Late Pleistocene (Roe, 2006; Bintanja and van de Wal, 2008; Abe-Ouchi et al., 2013). However, these mechanisms do not explain why large-scale NH glaciations only developed during the Late Pliocene and Pleistocene, and not during earlier time-periods with comparable astronomical configurations. The Pliocene-Pleistocene climatic-cryospheric evolution can therefore not be seen outside a context of long-term, multi-Myr changes in climatic and tectonic boundary conditions, not captured by the bispectral analysis presented here. For example, the "arc" in spectral power (~3,500–~2,000 ka) (Fig. 5c), which is largely absent in the bispectral results (Fig. 5b), and which precedes the strong glaciations of the MPT and Late Pleistocene, may be indicative of these gradually changing boundary conditions. We speculate that this arc reflects a change in the response time of the cryosphere, potentially through glacial landscaping by earlier (smaller) glacial cycles (Clark and Pollard, 1998; Tabor and Poulsen, 2016) or through changing tectonic/geographic boundary conditions and associated oceanic circulation patterns (Haug and Tiedemann, 1998; Kender et al., 2018). Such developments could have preconditioned the Earth's system for the full-blown bipolar glacial cycles of the Middle and Late Pleistocene. These cycles were amplified by variability in radiative forcing on astronomical time scales and set against a background of an all-Cenozoic-time low $P_{CO_2^{atm}}$ (Shackleton, 2000; Beerling and Royer, 2011; Martinez-Boti et al., 2015; Chalk et al., 2017).

## 6. Outlook

We present considerable advances in the understanding of the origins of many Pliocene and Pleistocene climate cycles by applying advanced bispectral analysis to the palaeoclimate archive. However, several lines of research remain to be explored further. Here we list those that we think need more attention in the future:

- For nearshore ocean waves, energy transfers are (almost) fully described by the imaginary part of the bispectrum (Norheim et al., 1998), because (*i*) nearshore ocean waves are dominated by asymmetric wave forms, and (*ii*) the imaginary part of bispectra in bispectral theory in general describe the majority of energy that is transferred. The importance of the imaginary part in describing most/all energy exchanges can easily be observed when computing power spectra of synthetically skewed and asymmetric time series (not shown here). Such spectra indicate that the higher harmonic peaks of skewed cycles are about an order of magnitude smaller than those of asymmetric cycles, especially for the higher harmonic peaks (3*f*, 4*f*, …) in comparison to the first higher harmonic (2*f*). Despite these

profound implications of bispectral theory, and the high asymmetry values of Middle and Late Pleistocene climate cycles, a more comprehensive understanding of the real part of the bispectrum may still be important for palaeoclimate studies, because it may enable an even more complete description of nonlinear energy exchanges (Supp. Fig. 1 and Supp. Fig. 2, e.g. see the ~28-kyr periodicity).

- Although we obtain conservative energy exchanges with our simplified, assumed coupling coefficient, additional studies are needed to scale these exchanges to the power spectrum (Fig. 6). Coupled climate-cryosphere models may be used to derive or approximate coupling coefficients for palaeoclimates that are based on "first principles" (i.e., a physicochemical understanding of the Earth system). Alternative coupling coefficients may also change how the bispectrum redistributes energy from climate cycles with low to those with high periodicities. However, we note that energy conservation should be respected for every triad interaction in the bispectrum.

- The proposed relationships between nonlinear triad interactions as documented in bispectra of climate cycles and specific processes of the climate-cryosphere system (i.e., monsoons, isostatic rebound), remain speculative at best. Therefore, we cannot currently separate these mechanisms from, for example, deep ocean carbon storage (e.g., (Willeit et al., 2015; Ganopolski et al., 2016; Farmer et al., 2019)), AMOC shutdown and subsequent overshoot (Liu et al., 2009; Wu et al., 2011), and/or changing sediment cover (Clark and Pollard, 1998; Ganopolski and Calov, 2011; Willeit et al., 2019) and the amplifying effects these processes may have had on the strength of the 40 and ~110 kyr cycles. More data (analysis) and/or (isotope-enabled) modelling studies are needed to support our preferred hypotheses. For example, global climate models could potentially help identify which mechanisms correspond to the specific frequency interactions we observe in the LR04 stack, and energy balance models may provide further insights in the amounts of energy that are transferred.

- There is a limited amount of information that can be extracted from studying a single climate record. Bispectral analysis of other (Pliocene-Pleistocene) records may shed more light on the dynamics of Earth's palaeoclimate system. Not only in elucidating the couplings among astronomical frequencies, but also across the spectral continuum, in relation to $1/f$ "red" noise, and/or the power-law scaling of energy exchanges between frequencies (see Section 2.4.1. and Section 4.2) (Hasselmann, 1976; Bak et al., 1987, 1988; Bak, 1996; Pelletier, 1998; Huybers and Curry, 2006). Examples of interactions that could be studied in this way are those between astronomical and sub-astronomical (e.g., millennial) climate cycles (Hagelberg et al., 1994; von Dobeneck and Schmieder, 1999; Wara et al., 2000; Da Silva et al., 2018), and among cycles with centennial, decadal and perhaps even annual durations (King Hagelberg and Cole, 1995). We emphasize that other applications of bispectral analysis techniques to the palaeoclimate archive should be limited to records with good age control and high signal-to-noise ratios (see Section 4.1.1).

# 7. Conclusions

In this interdisciplinary study, we present a new, higher-order spectral analysis and interpretation of Pliocene and Pleistocene climate cycles as present in the well-established LR04 globally-averaged benthic-foraminiferal $\delta^{18}O$ stack. We use the nonsinusoidal properties of these climate cycles as a proxy for the strength of nonlinear couplings and interactions among different components of the climate system that operate on distinctly separate time scales. These advanced bispectral analysis techniques were initially developed by the nearshore ocean wave community, and are adapted here to make them more applicable for paleoclimatic research purposes. They show, more thoroughly than before, the complexities of the Pliocene-Pleistocene climatic-cryospheric evolution, namely that (*i*) a nonlinear energy exchanges resulted in the distinct asymmetric cycle geometries of the Middle and Late Pleistocene, (*ii*) energy is transferred across the power spectrum, among astronomical and nonastronomical climate cycles, (*iii*) precession-paced climate cycles fuelled both the 40-kyr and (largely indirectly) the ~110-kyr ice age cycles of the Pliocene and Pleistocene, (*iv*) the role of obliquity-paced climate cycles changes from being a net sink into a net source of energy during the MPT (i.e., in $B^{Im}$(E, E, O)-interactions), and (*v*) after the MPT these obliquity-paced climate cycles helped fuel the ~110-kyr eccentricity-paced climate cycles of the Middle and Late Pleistocene.

We postulate two distinct fuelling mechanisms for the ice ages to explain the energy cascade of negative interactions from climate cycles with high frequencies to those with low frequencies: (*i*) a continuous Pliocene-Pleistocene climatic "precession motor" and (*ii*) a Middle-to-Late Pleistocene cryospheric "obliquity motor". We interpret the dominant precession fuelling of obliquity paced climate cycles (i.e., the precession motor) as evidence of a low latitude, (sub-)tropically-driven climate system, in which the monsoons (and oceanic meridional overturning circulation) transport heat and moisture to higher latitudes. We argue that the role-reversal of obliquity-paced climate cycles during the MPT is indicative of the passing of a land-ice mass-loading threshold, after which autocycles of crustal sinking and (delayed) rebound start to resonate, i.e., become frequency- and phase-coupled, with the ~110-kyr long allocycles of eccentricity-modulated precession. This resonance may have caused a shift of sensitivity to (obliquity-paced) insolation changes at increasingly higher latitudes, especially during terminations.

Despite the fact that the evidence for energy transfers agrees well with previously published mechanisms, their unprecedentedly detailed description here is an important step forward toward a more comprehensive solution of the problem of the ice ages, because they largely reconcile the mismatch in power spectral density between records that capture the combined effects of deep-sea temperatures and land-ice volumes and those of insolation. Furthermore, we can now state with greater certainty than before, that the asymmetry of the Pliocene and Pleistocene ice ages is in very close agreement with Neo-Milankovitchism: i.e., the classical Milankovitch Theory of astronomical climate forcing and its Neo-Milankovitchistic derivatives/superlatives that give greater weight to Earth's internal nonlinear responses. If the bispectral energy transfers

documented here indeed track the entire redistribution of power over the spectrum, then the 40-kyr problem of the Pliocene and Early Pleistocene, and the ~110-kyr problem of the Middle and Late Pleistocene are reduced to finding appropriate coupling coefficients that describe the relevant climatic and cryospheric mechanisms–a promising outlook.

## 5  Acknowledgments

We thank X. Bertin for inviting D. L. to present this work at an early stage at the Université de La Rochelle. We are grateful to T. E. van Peer for making us aware of the difficulties in quantifying robust geometries of nonergodic records. We thank L. E. Lisiecki and M. E. Raymo, and S. Ahn et al. for making their benthic foraminiferal $\delta^{18}O$ stacks available online. We thank T. Herbers and G. Ruessink for providing bispectral analysis MATLAB scripts, which A. T. M. d. B. developed further for the purposes of this study.

**Figure 1.** Sinusoidal properties of Pliocene and Pleistocene climate cycles (after (Lisiecki and Raymo, 2005; Lisiecki, 2010)). **(a)** Resampled LR04 benthic foraminiferal $\delta^{18}O$ stack. **(b)** Wavelet analysis of the LR04 stack after detrending. Black contours represent 95% significance. **(c)** Absolute amplitude modulations and filters of the ~110-kyr, 40 kyr, and ~21-kyr periodicities. **(d)** Phase evolution of the ~110-kyr periodicity with respect to eccentricity (Laskar et al., 2011a; Laskar et al., 2011b), computed over a 750 kyr sliding window. Blackman-Tukey phase estimates are only shown when coherent (Paillard et al., 1996). We focus the phase interpretation on the time interval with high amplitude ~110-kyr long cycles (transparent area). Phase relationship of $\delta^{18}O$ to obliquity is not shown because it is assumed during the astronomical tuning process. Phase to precession is not shown because globally averaged $\delta^{18}O$ records have no single response to either NH or SH precession forcing. **(e)** Obliquity, eccentricity and precession solutions. The 1.2-Myr obliquity, and 2.4-Myr and 405-kyr eccentricity cycles numbers are given. Black and white bars correspond to normal and reversed magnetochrons of the 2012 geologic time scale (Hilgen et al., 2012).

**Figure 2.** Nonsinusoidal properties of Pliocene and Pleistocene climate cycles (after (Lisiecki and Raymo, 2007)). **(a)** As in Figure 1. **(b)** Evolution of glacial-interglacial cycle skewness, **(c)** asymmetry, and **(d)** excess kurtosis (i.e., normalised third and fourth central moments). Computed over a 668 kyr sliding window, using standard (teal and blue lines) and bispectral (pink lines) methods. Cycle kurtosis is computed with the standard method only, because no trispectra were computed. Inset boxes show corresponding cycle shapes. The arrow shows the direction of time, g = glacial, ig = interglacial. We focus the geometry interpretation on the time interval with the greater amplitude variability in the record (transparent area). **(e)** As in Figure 1.

**Figure 3.** Bispectral zonation scheme. Bispectra of climate cycles can be subdivided into 15 zones that reflect unique combinations of three frequency bandwidths. Orange, purple, and blue lines represent the main astronomical frequencies of the eccentricity, obliquity and precession cycles, respectively (this colour coding is consistent throughout the paper). The grey shaded areas represent suborbital periodicities. Difference frequencies $f_1$ and $f_2$ are plotted along the along the horizontal and vertical axes, respectively. Sum frequencies $f_3$ (i.e., $f_1 + f_2$) are depicted as diagonal lines in the bispectrum. See also Table A1 for a list of bispectral zones, and Table A2 for a list of triad types among the main astronomical frequencies in the bispectrum.

**Figure 4.** Bispectra of Pliocene and Pleistocene climate cycles. **(a)** Imaginary part of a bispectrum of the Middle and Late Pleistocene "~110-kyr world". Age range = 668 ka to 0 ka. Resampling resolution = 1 kyr$^{-1}$. Window length = 668 kyr. Block length = 668 kyr. Number of blocks = 1. Number of merged frequencies = 1. Frequency resolution = 1.50 Myr$^{-1}$. Degrees of freedom = 2. Min value = −598 ‰$^3$ kyr$^2$. Max value = 51 ‰$^3$ kyr$^2$. **(b)** Imaginary part of a bispectrum of the mid-Pleistocene transition "~80-kyr world". Age range = 1,318 ka to 650 ka. Resampling resolution = 1 kyr$^{-1}$. Window length = 668 kyr. Block length = 668 kyr. Number of blocks = 1. Number of merged frequencies = 1. Frequency resolution = 1.50 Myr$^{-1}$. Degrees of freedom = 2. Min value = −57 ‰$^3$ kyr$^2$. Max value = 73 ‰$^3$ kyr$^2$. **(c)** Imaginary part of a bispectrum of the Pliocene and Early Pleistocene "40-kyr world". Age range = 3,000 ka to 1,000 ka. Resampling resolution = 1 kyr$^{-1}$. Window length = 2,000 kyr. Block length = 2,000 kyr. Number of blocks = 1. Number of merged frequencies = 3. Frequency resolution = 1.50 Myr$^{-1}$. Degrees of freedom = 6. Min value = −21 ‰$^3$ kyr$^2$. Max value = 23 ‰$^3$ kyr$^2$. Note the different scaling of the z-axes between panel (a), (b), and (c). These bispectra are computed on the LR04 stack (see Methods) (Lisiecki and Raymo, 2005).

**Figure 5. (a)** Input $\rightarrow$ **(b)** "black box" climate $\rightarrow$ **(c)** output. **(a)** Power spectral density of mean summer insolation (June 21 to September 21) at 65°N (Paillard et al., 1996; Laskar et al., 2004). **(b)** Conservative net energy transfers (i.e., after correction using the coupling coefficient) during the Pliocene and Pleistocene, computed by integrating over the entire imaginary part of the bispectrum (see Methods). Input data is the resampled LR04 stack (Lisiecki and Raymo, 2005). **(c)** Power spectral density of the LR04 stack (Lisiecki and Raymo, 2005). For all three panels, the window length is 668 data points (668 kyr), and the step-size between partially overlapping windows is 50 data points (50 kyr). We used no blocks and did not merge frequencies. These settings yield a frequency resolution of 1.50 Myr$^{-1}$, and 2 degrees of freedom.

**Figure 6.** Qualitative linkage of power spectra of insolation to those of climate, via energy exchanges among climate cycles. (**a**) 334 ka. (**b**) 834 ka. (**c**) 1334 ka. (**d**) 1834 ka. (**e**) 2334 ka. Computational settings as in Figure 5. Note that the scaling between the spectra and energy exchanges is arbitrary. Energy exchanges of the LR04 stack can be visually "scaled" (i.e., qualitatively compared) to the power spectrum of the LR04 by using multiplication factors of 430, 250, 360, 230, and 250 for panels (a) to (e), respectively.

**Figure 7.** Integration of the spectra and bispectra per astronomical bandwidth (top right). (**a**) Totally and zonally integrated spectra of mean summer insolation (June 21 to September 21) at 65°N. (**b**) Totally and zonally integrated bispectra of the LR04 stack. (**c**) Totally and zonally integrated spectra of the LR04 stack. Computational settings as in Figure 5.

**Figure 8.** Disentangling "black box" climate. Conservative net energy transfers during the Pliocene and Pleistocene over specific zones in the imaginary part of the bispectrum (see Methods). Computational settings as in Figure 5b. (**a**) Zone 1. (**b**) Zone 2. (**c**) Zone 3. (**d**) Zone 4. (**e**) Zone 5. (**f**) Zone 6. (**g**) Zone 7. (**h**) Zone 8. See also Figure 3 and Table A1.

**Figure 9.** Conservation of energy in triad interactions located in the imaginary part of Pliocene and Pleistocene bispectra. Conservativity of (**a**) the entire bispectrum (Fig. 5b), and (**b**) Zone 1, (**c**) Zone 2, (**d**) Zone 3, (**e**) Zone 4, (**f**) Zone 5, (**g**) Zone 6, (**h**) Zone 7, and (**i**) Zone 8, of the bispectrum (Fig. 8). Energy gains or losses of $f_1$ and $f_2$ (dashed lines) are computed using Eq. 8, those of $f_3$ (single coloured lines) following Eq. 7, and their differences (black lines) by Eq. 6.

**Figure 10.** Summed zonal integrations of the imaginary part of Pliocene and Pleistocene bispectra. Net energy transfers are computed by (**a**) summing Zones 1, 2, 3, 4, and 6 for triad interactions corresponding to the "eccentricity" bandwidth, (**b**) summing Zones 2, 3, 4, 5, and 7 for triad interactions corresponding to the "obliquity" bandwidth, and (**c**) summing Zones 4, 5, 6, 7, and 8 for triad interactions corresponding to the "precession" bandwidth. Panel (**d**) shows the corresponding summed zonal conservation of energy, with eccentricity in orange, obliquity in purple, and precession in blue.

**Appendices**

**Table A1.** Bispectral zones defined in this study. These zones correspond to Fig. 3. The boundaries between the climate cycle zones are (arbitrarily) defined at periodicities of ~56.2, ~28.1 and 12.5 kyr (i.e., frequencies of 17.8, 35.6 and 80.0 $Myr^{-1}$).

**Table A2.** Bispectral triads among climate cycles with astronomical frequencies. One, two, or three astronomical frequencies can partake in triad interactions, and we refer to these coordinates in the bispectrum as single, double or triple junctions, respectively. These triads reflect positions in the frequency-frequency domain where nonlinear energy transfers among astronomically-paced climate cycles are likely to occur. They correspond to the crossing points of the coloured lines in Fig. 3. In case of a "single junction", an astronomically-paced cycle exchanges energy with itself, or with a frequency that is only slightly offset. Single junctions where $f_1 = f_2$ constitute the first higher harmonic interactions. Double or triple junctions are "combination tones" of two or three different astronomically paced cycles (Pestiaux et al., 1988; Hagelberg et al., 1991; King, 1996; Rial and Anaclerio, 2000).

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

# Liebrand & de Bakker, Figure 1, Column width 1

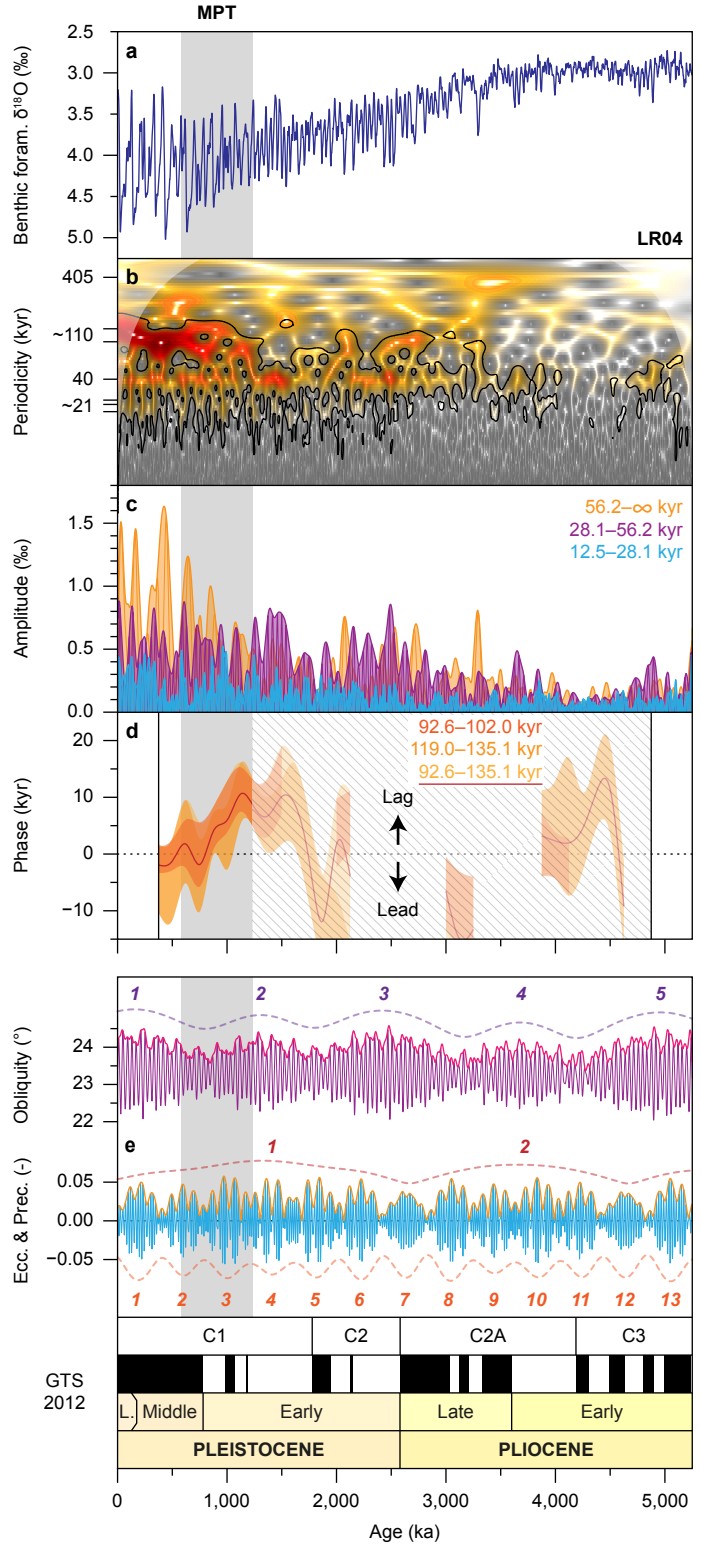

# Liebrand & de Bakker, Figure 2, Column width 1

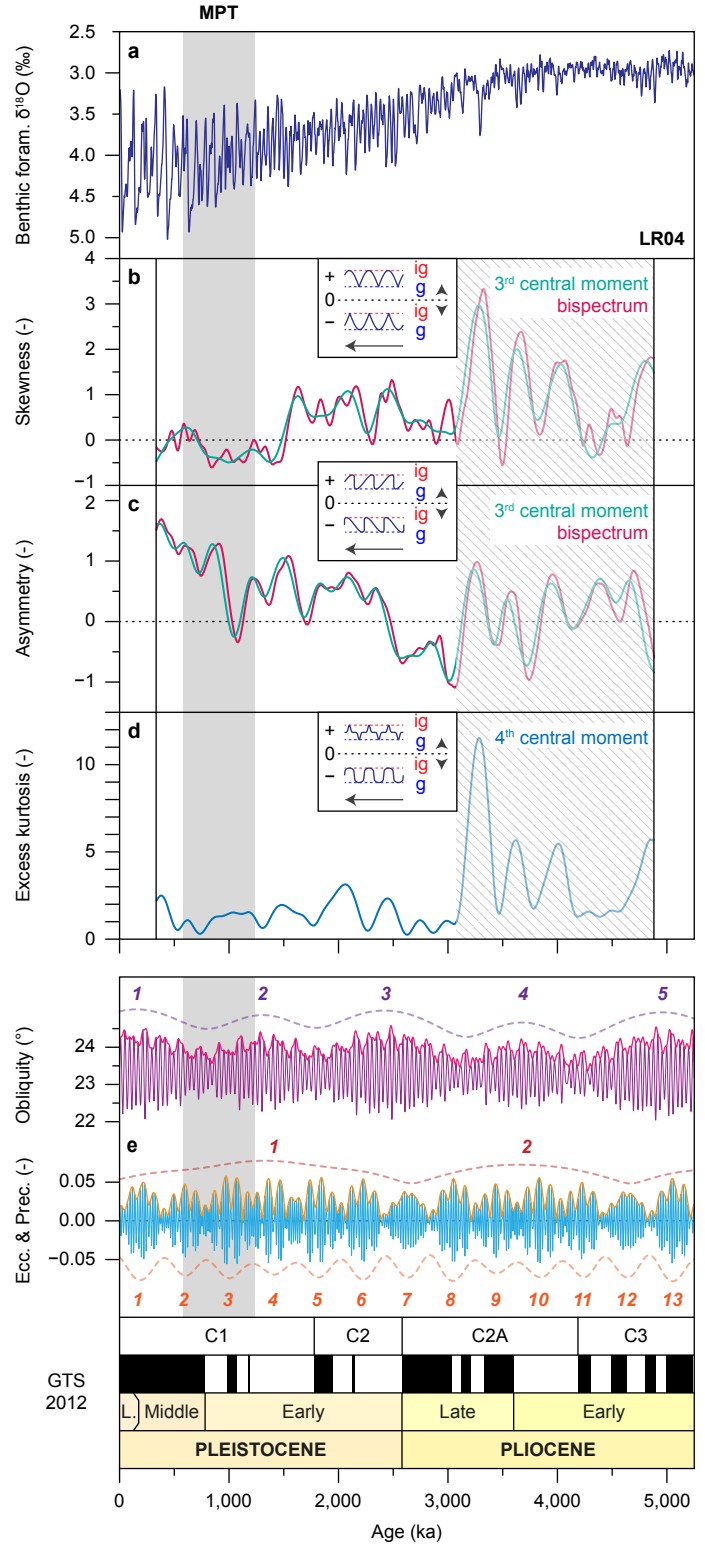

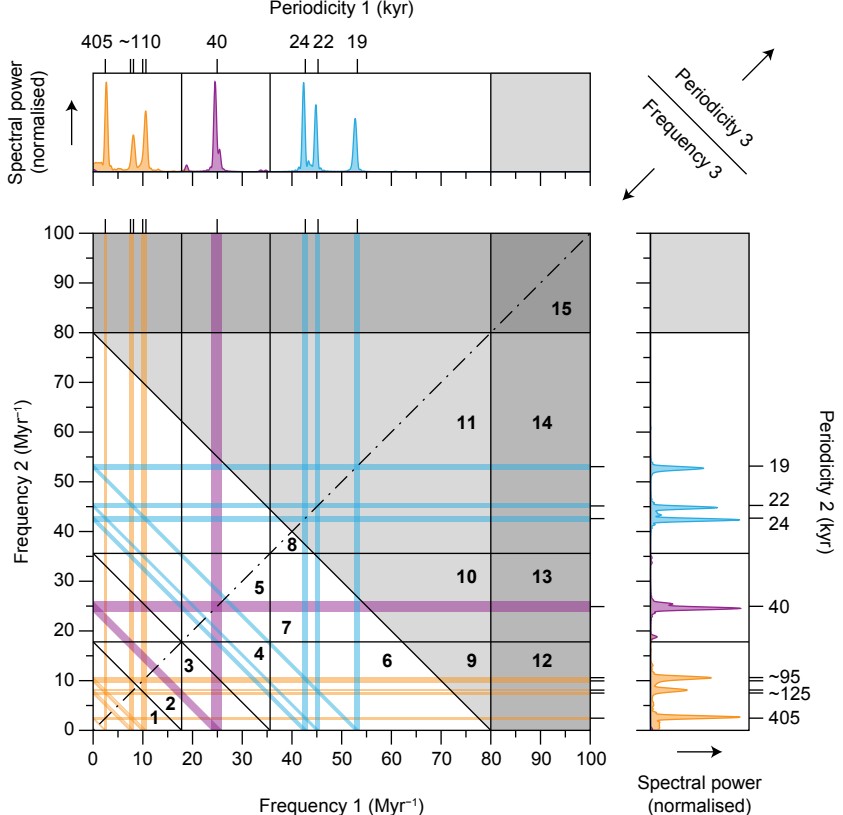

**Liebrand & de Bakker, Figure 4, Column width 1**

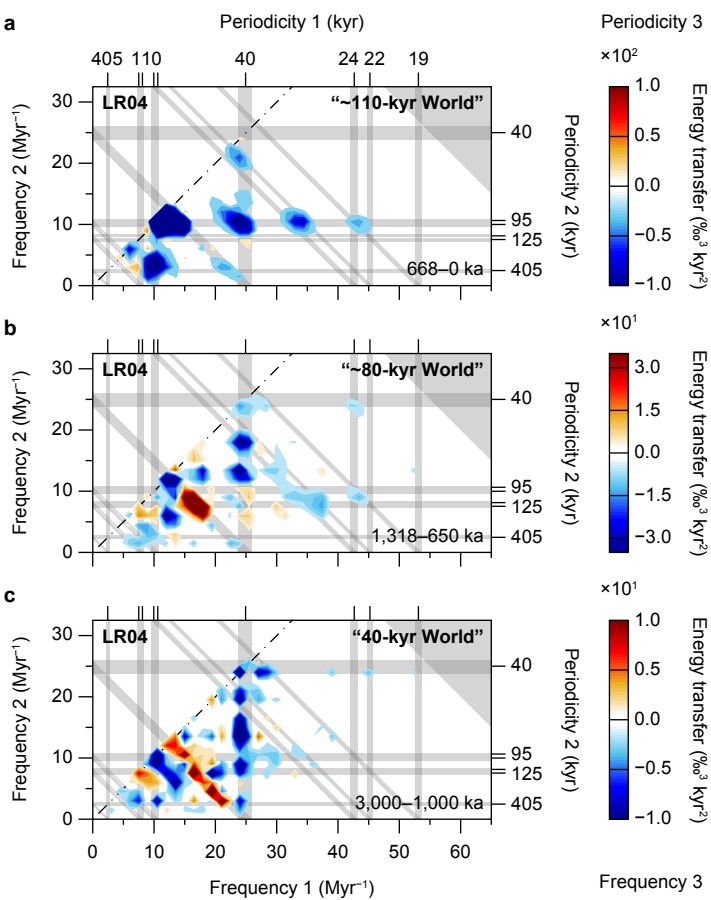

# Liebrand & de Bakker, Figure 5, Column width 1

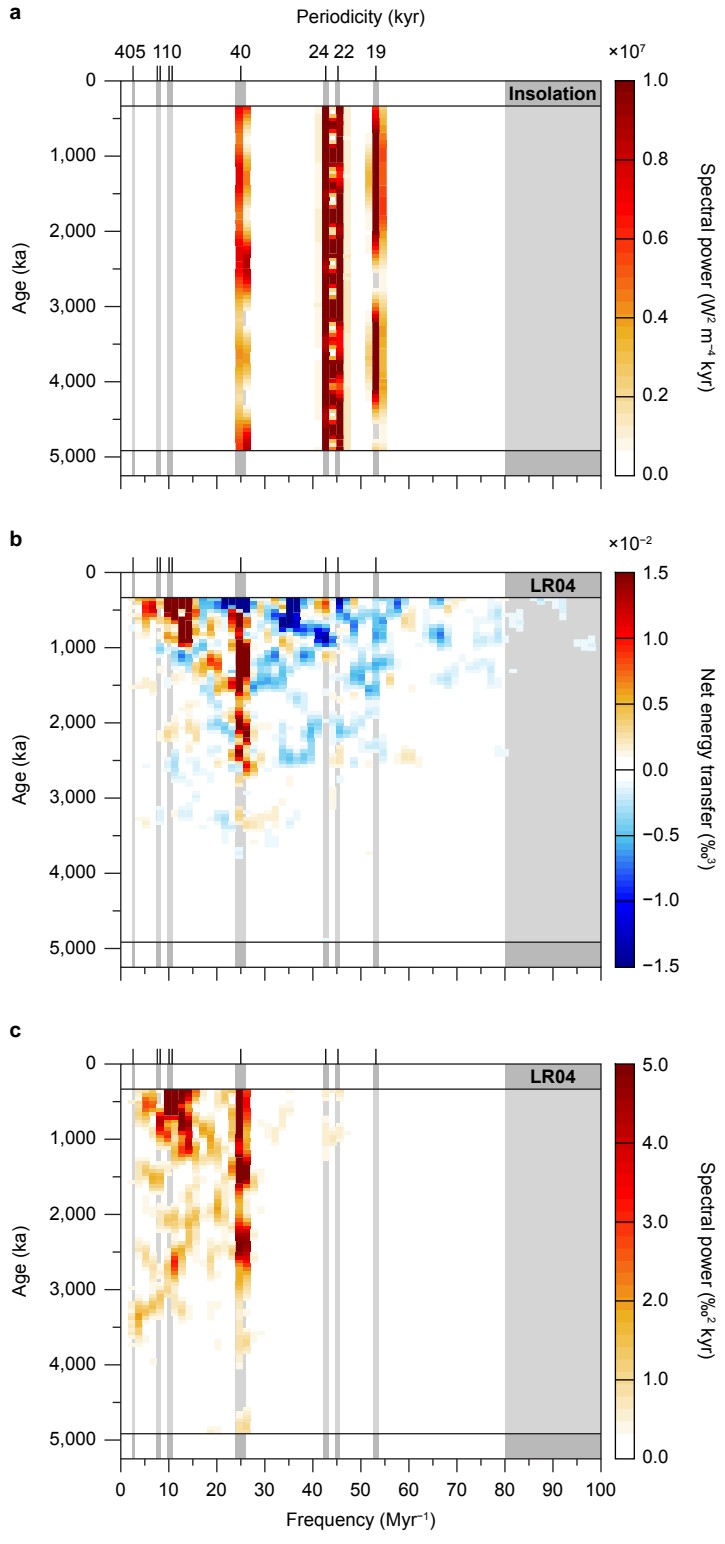

**Liebrand & de Bakker, Figure 6, Column width 1**

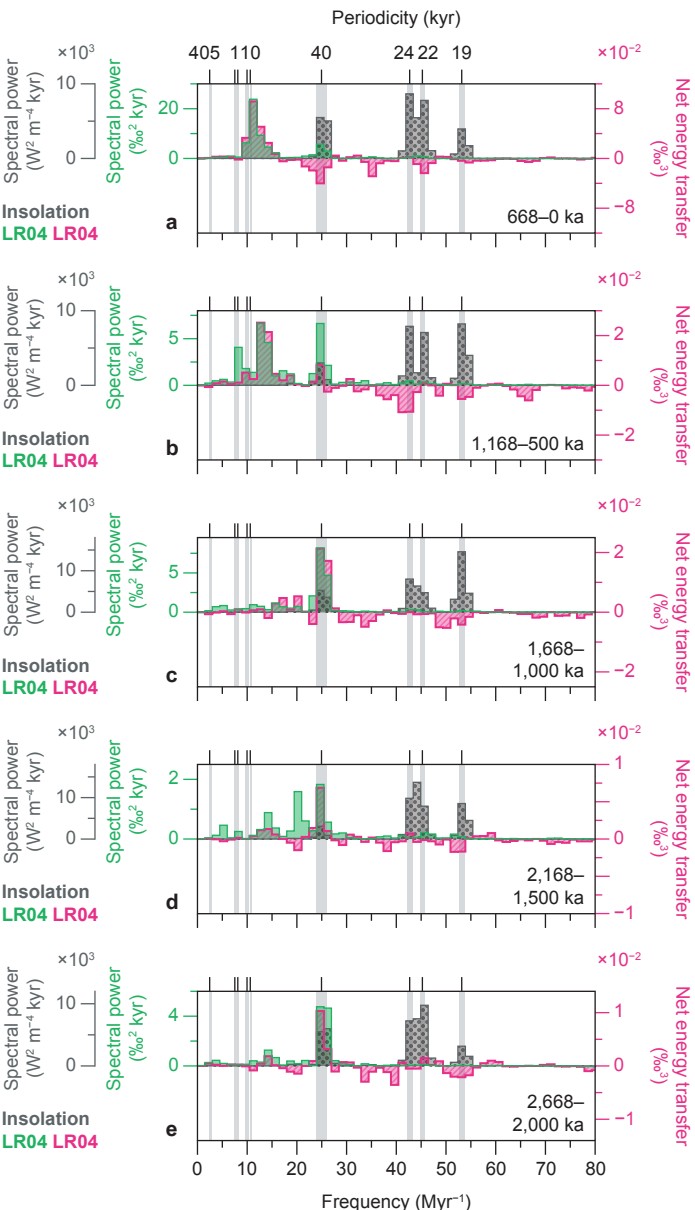

**Liebrand & de Bakker, Figure 7, Column width 1**

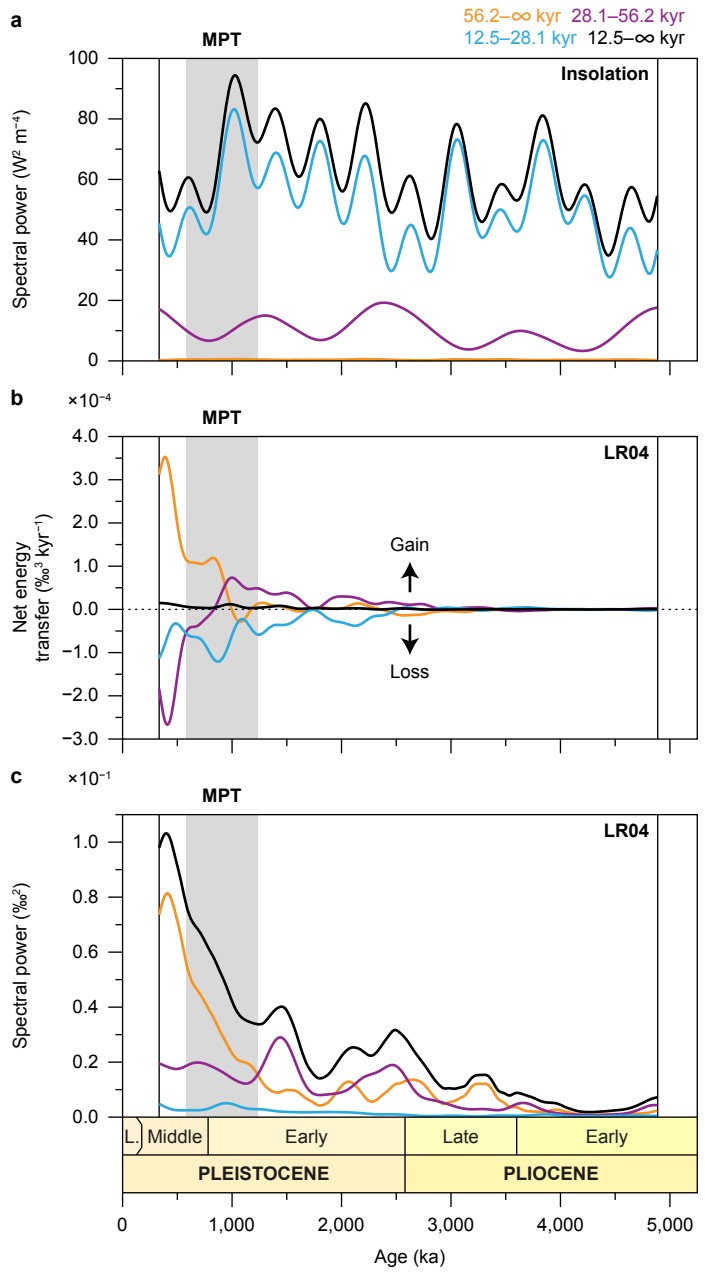

# Liebrand & de Bakker, Figure 8, Column width 2

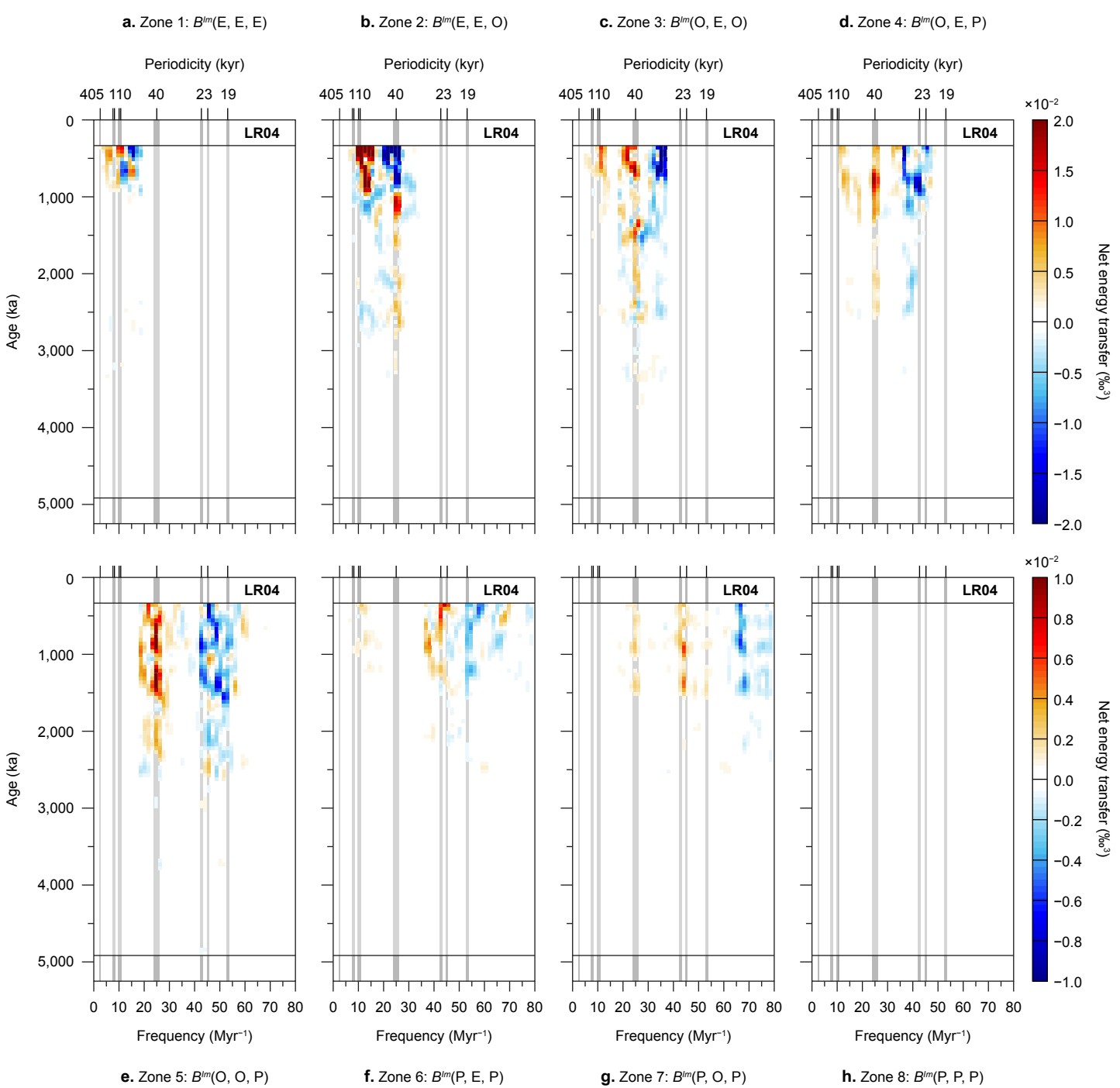

**a.** Zone 1: $B^{lm}$(E, E, E)  **b.** Zone 2: $B^{lm}$(E, E, O)  **c.** Zone 3: $B^{lm}$(O, E, O)  **d.** Zone 4: $B^{lm}$(O, E, P)

**e.** Zone 5: $B^{lm}$(O, O, P)  **f.** Zone 6: $B^{lm}$(P, E, P)  **g.** Zone 7: $B^{lm}$(P, O, P)  **h.** Zone 8: $B^{lm}$(P, P, P)

**Liebrand & de Bakker, Figure 9, Column width 2**

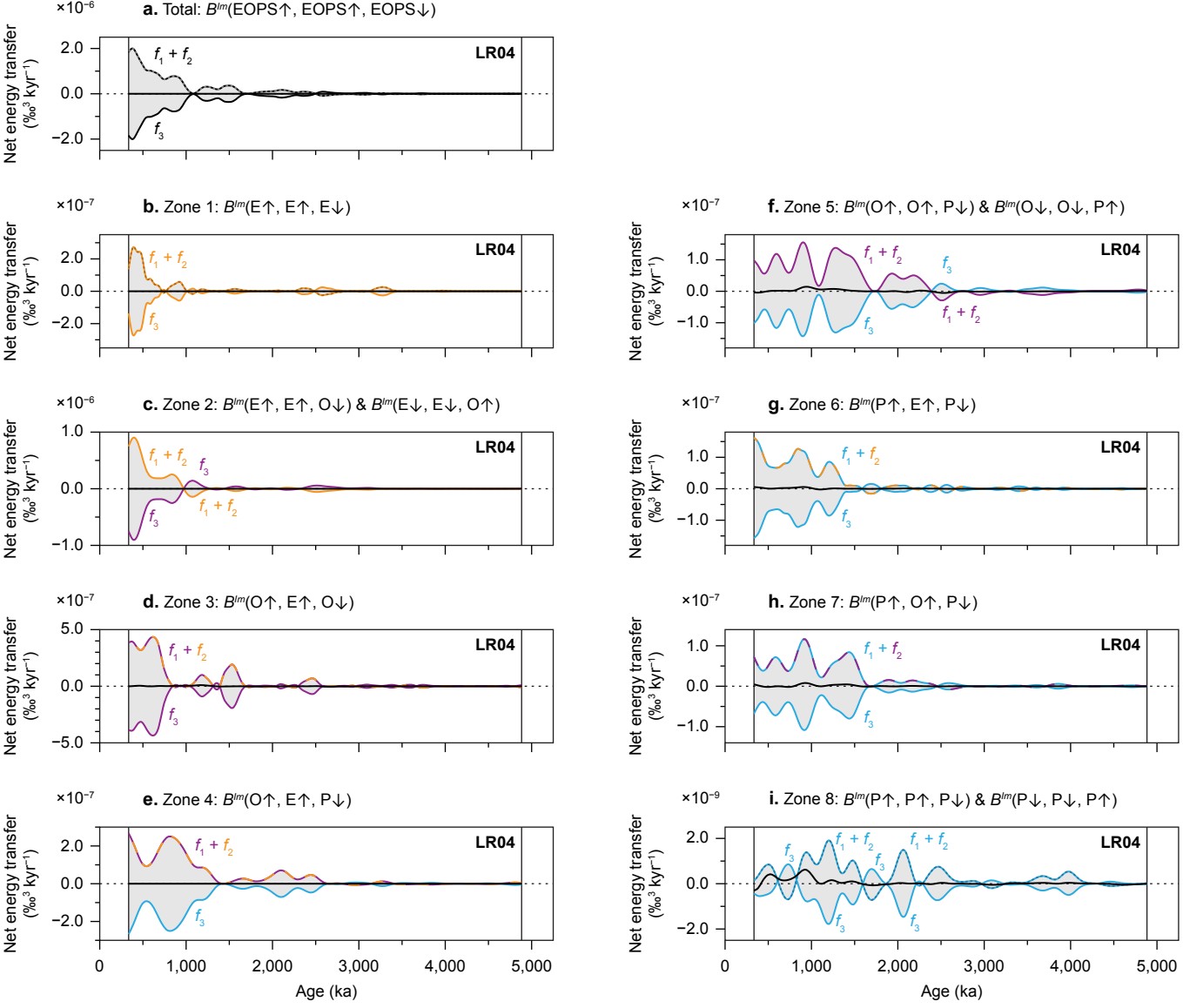

**Liebrand & de Bakker, Figure 10, Column width 2**

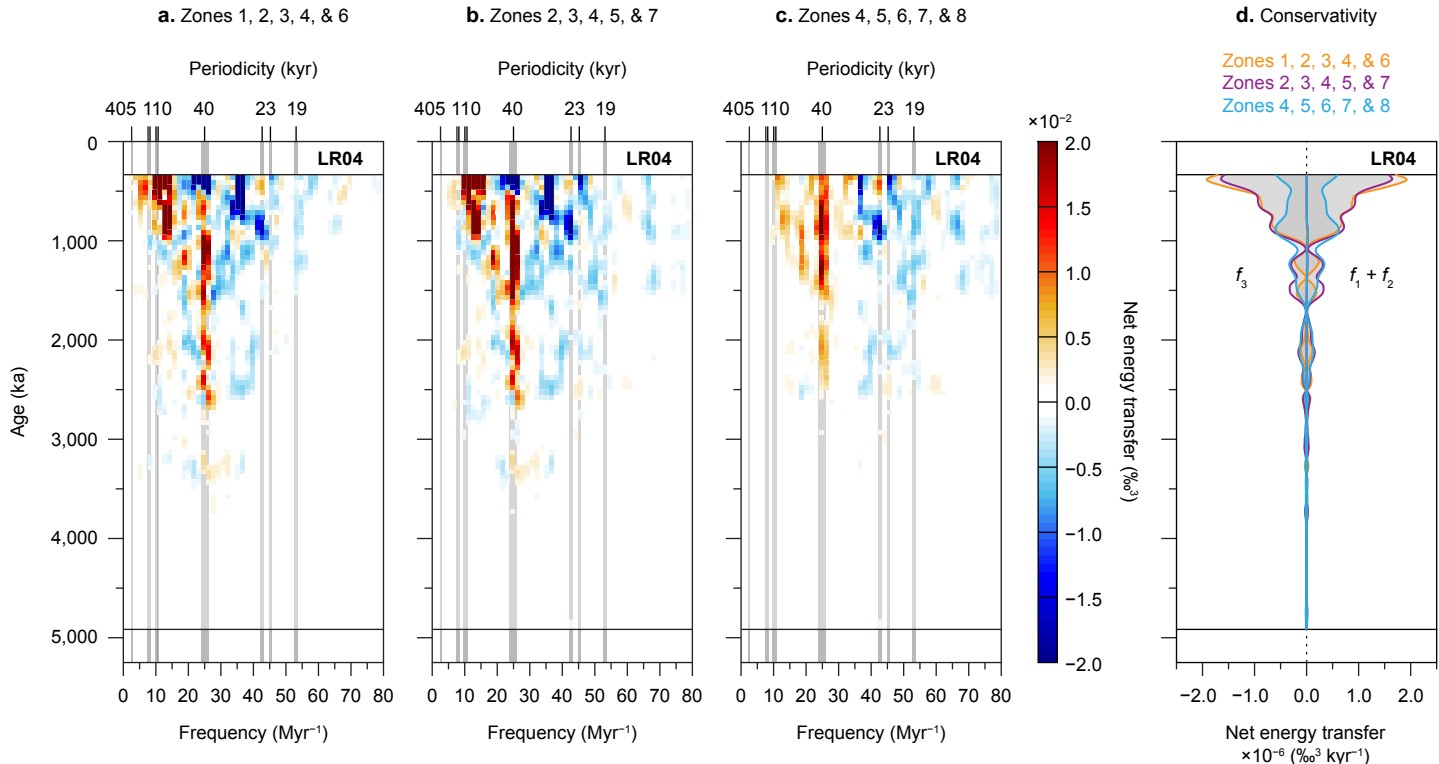

**Liebrand & de Bakker, Appendix: Table A1**

| Bispectral zone | Periodicity 1 (difference) | Periodicity 2 (difference) | Periodicity 3 (sum) | Bispectral notation |
|---|---|---|---|---|
| 1 | Eccentricity | Eccentricity | Eccentricity | $B$(E, E, E) |
| 2 | Eccentricity | Eccentricity | Obliquity | $B$(E, E, O) |
| No zone | Eccentricity | Eccentricity | Precession | $B$(E, E, P) |
| No zone | Eccentricity | Eccentricity | Suborbital | $B$(E, E, S) |
| 3 | Obliquity | Eccentricity | Obliquity | $B$(O, E, O) |
| 4 | Obliquity | Eccentricity | Precession | $B$(O, E, P) |
| No zone | Obliquity | Eccentricity | Suborbital | $B$(O, E, S) |
| No zone | Obliquity | Obliquity | Obliquity | $B$(O, O, O) |
| 5 | Obliquity | Obliquity | Precession | $B$(O, O, P) |
| No zone | Obliquity | Obliquity | Suborbital | $B$(O, O, S) |
| 6 | Precession | Eccentricity | Precession | $B$(P, E, P) |
| 7 | Precession | Obliquity | Precession | $B$(P, O, P) |
| 8 | Precession | Precession | Precession | $B$(P, P, P) |
| 9 | Precession | Eccentricity | Suborbital | $B$(P, E, S) |
| 10 | Precession | Obliquity | Suborbital | $B$(P, O, S) |
| 11 | Precession | Precession | Suborbital | $B$(P, P, S) |
| 12 | Suborbital | Eccentricity | Suborbital | $B$(S, E, S) |
| 13 | Suborbital | Obliquity | Suborbital | $B$(S, O, S) |
| 14 | Suborbital | Precession | Suborbital | $B$(S, P, S) |
| 15 | Suborbital | Suborbital | Suborbital | $B$(S, S, S) |

# Liebrand & de Bakker, Appendix: Table A2

| Bispectral zone(s) | $B_f(f_1, f_2, f_3)$ (Myr$^{-1}$) | $B_p(p_1, p_2, p_3)$ (kyr) | Triad type |
|---|---|---|---|
| 1 | $B_f(\mathbf{2.5}, 0.0, \mathbf{2.5})$ | $B_p(\mathbf{405}, \infty, \mathbf{405})$ | Single junction |
| 1 | $B_f(\mathbf{2.5}, \mathbf{2.5}, 4.9)$ | $B_p(\mathbf{405}, \mathbf{405}, 203)$ | Single junction |
| 1 | $B_f(5.5, \mathbf{2.5}, \mathbf{8.0})$ | $B_p(181, \mathbf{405}, \mathbf{125})$ | Double junction |
| 1 | $B_f(\mathbf{8.0}, 0.0, \mathbf{8.0})$ | $B_p(\mathbf{125}, \infty, \mathbf{125})$ | Single junction |
| 1 | $B_f(\mathbf{8.0}, \mathbf{2.5}, \mathbf{10.5})$ | $B_p(\mathbf{125}, \mathbf{405}, \mathbf{95})$ | Triple junction |
| 1 | $B_f(\mathbf{8.0}, \mathbf{8.0}, 16.0)$ | $B_p(\mathbf{125}, \mathbf{125}, 63)$ | Single junction |
| 1 | $B_f(\mathbf{10.5}, 0.0, \mathbf{10.5})$ | $B_p(\mathbf{95}, \infty, \mathbf{95})$ | Single junction |
| 1 | $B_f(\mathbf{10.5}, \mathbf{2.5}, 13.0)$ | $B_p(\mathbf{95}, \mathbf{405}, 77)$ | Double junction |
| 1, 2 | $B_f(\mathbf{10.5}, \mathbf{8.0}, 18.5)$ | $B_p(\mathbf{95}, \mathbf{125}, 54)$ | Double junction |
| 2 | $B_f(\mathbf{10.5}, \mathbf{10.5}, 21.0)$ | $B_p(\mathbf{95}, \mathbf{95}, 48)$ | Single junction |
| 2 | $B_f(14.5, \mathbf{10.5}, \mathbf{25.0})$ | $B_p(69, \mathbf{95}, \mathbf{40})$ | Double junction |
| 2, (3) | $B_f(17.0, \mathbf{8.0}, \mathbf{25.0})$ | $B_p(59, \mathbf{125}, \mathbf{40})$ | Double junction |
| 3 | $B_f(22.5, \mathbf{2.5}, \mathbf{25.0})$ | $B_p(44, \mathbf{405}, \mathbf{40})$ | Double junction |
| 3 | $B_f(\mathbf{25.0}, 0.0, \mathbf{25.0})$ | $B_p(\mathbf{40}, \infty, \mathbf{40})$ | Single junction |
| 3 | $B_f(\mathbf{25.0}, \mathbf{2.5}, 27.5)$ | $B_p(\mathbf{40}, \mathbf{405}, 36)$ | Double junction |
| 3 | $B_f(\mathbf{25.0}, \mathbf{8.0}, 33.0)$ | $B_p(\mathbf{40}, \mathbf{125}, 30)$ | Double junction |
| 3, 4 | $B_f(\mathbf{25.0}, \mathbf{10.5}, 35.5)$ | $B_p(\mathbf{40}, \mathbf{95}, 28)$ | Double junction |
| 4, 5 | $B_f(\mathbf{25.0}, 17.3, \mathbf{42.3})$ | $B_p(\mathbf{40}, 58, \mathbf{24})$ | Double junction |
| 5 | $B_f(\mathbf{25.0}, 19.8, \mathbf{44.8})$ | $B_p(\mathbf{40}, 51, \mathbf{22})$ | Double junction |
| 5 | $B_f(\mathbf{25.0}, \mathbf{25.0}, 50.0)$ | $B_p(\mathbf{40}, \mathbf{40}, 20)$ | Single junction |
| 5 | $B_f(27.8, \mathbf{25.0}, 52.8)$ | $B_p(36, \mathbf{40}, \mathbf{19})$ | Double junction |
| 4 | $B_f(31.8, \mathbf{10.5}, \mathbf{42.3})$ | $B_p(31, \mathbf{95}, \mathbf{24})$ | Double junction |
| 4 | $B_f(34.3, \mathbf{8.0}, \mathbf{42.3})$ | $B_p(29, \mathbf{125}, \mathbf{24})$ | Double junction |
| 4 | $B_f(34.3, \mathbf{10.5}, \mathbf{44.8})$ | $B_p(29, \mathbf{95}, \mathbf{22})$ | Double junction |
| 6 | $B_f(36.8, \mathbf{8.0}, \mathbf{44.8})$ | $B_p(27, \mathbf{125}, \mathbf{22})$ | Double junction |
| 6 | $B_f(39.8, \mathbf{2.5}, \mathbf{42.3})$ | $B_p(25, \mathbf{405}, \mathbf{24})$ | Double junction |
| 6 | $B_f(\mathbf{42.3}, 0.0, \mathbf{42.3})$ | $B_p(\mathbf{24}, \infty, \mathbf{24})$ | Single junction |
| 6 | $B_f(\mathbf{42.3}, \mathbf{2.5}, \mathbf{44.8})$ | $B_p(\mathbf{24}, \mathbf{405}, \mathbf{22})$ | Triple junction |
| 6 | $B_f(\mathbf{42.3}, \mathbf{8.0}, 50.3)$ | $B_p(\mathbf{24}, \mathbf{125}, 20)$ | Double junction |
| 6 | $B_f(\mathbf{42.3}, \mathbf{10.5}, \mathbf{52.8})$ | $B_p(\mathbf{24}, \mathbf{95}, \mathbf{19})$ | Triple junction |
| 7 | $B_f(\mathbf{42.3}, \mathbf{25.0}, 67.3)$ | $B_p(\mathbf{24}, \mathbf{40}, 15)$ | Double junction |
| 11 | $B_f(\mathbf{42.3}, \mathbf{42.3}, 84.6)$ | $B_p(\mathbf{24}, \mathbf{24}, 12)$ | Single junction |
| 6 | $B_f(\mathbf{44.8}, 0.0, \mathbf{44.8})$ | $B_p(\mathbf{22}, \infty, \mathbf{22})$ | Single junction |
| 6 | $B_f(\mathbf{44.8}, \mathbf{2.5}, 47.3)$ | $B_p(\mathbf{22}, \mathbf{405}, 21)$ | Double junction |
| 6 | $B_f(\mathbf{44.8}, \mathbf{8.0}, \mathbf{52.8})$ | $B_p(\mathbf{22}, \mathbf{125}, \mathbf{19})$ | Triple junction |
| 6 | $B_f(\mathbf{44.8}, \mathbf{10.5}, 55.3)$ | $B_p(\mathbf{22}, \mathbf{95}, 18)$ | Double junction |
| 7 | $B_f(\mathbf{44.8}, \mathbf{25.0}, 69.8)$ | $B_p(\mathbf{22}, \mathbf{40}, 14)$ | Double junction |
| 11 | $B_f(\mathbf{44.8}, \mathbf{42.3}, 87.1)$ | $B_p(\mathbf{22}, \mathbf{24}, 11)$ | Double junction |
| 11 | $B_f(\mathbf{44.8}, \mathbf{44.8}, 89.6)$ | $B_p(\mathbf{22}, \mathbf{22}, 11)$ | Single junction |
| 6 | $B_f(50.3, \mathbf{2.5}, \mathbf{52.8})$ | $B_p(20, \mathbf{405}, \mathbf{19})$ | Double junction |
| 6 | $B_f(\mathbf{52.8}, 0.0, \mathbf{52.8})$ | $B_p(\mathbf{19}, \infty, \mathbf{19})$ | Single junction |
| 6 | $B_f(\mathbf{52.8}, \mathbf{2.5}, 55.3)$ | $B_p(\mathbf{19}, \mathbf{405}, 18)$ | Double junction |
| 6 | $B_f(\mathbf{52.8}, \mathbf{8.0}, 60.8)$ | $B_p(\mathbf{19}, \mathbf{125}, 16)$ | Double junction |
| 6 | $B_f(\mathbf{52.8}, \mathbf{10.5}, 63.3)$ | $B_p(\mathbf{19}, \mathbf{95}, 16)$ | Double junction |
| 7 | $B_f(\mathbf{52.8}, \mathbf{25.0}, 77.8)$ | $B_p(\mathbf{19}, \mathbf{40}, 13)$ | Double junction |
| 11 | $B_f(\mathbf{52.8}, \mathbf{42.3}, 95.1)$ | $B_p(\mathbf{19}, \mathbf{24}, 11)$ | Double junction |
| 11 | $B_f(\mathbf{52.8}, \mathbf{44.8}, 97.6)$ | $B_p(\mathbf{19}, \mathbf{22}, 10)$ | Double junction |
| 11 | $B_f(\mathbf{52.8}, \mathbf{52.8}, 105.6)$ | $B_p(\mathbf{19}, \mathbf{19}, 9)$ | Single junction |