# Peer review of "Bispectra of climate cycles show how ice ages are fuelled"

_Climate of the Past, 2019_

## Referee Comment (RC1) · Michel Crucifix (Referee) · 14 May 2019

**1  Summary**

The authors present an extensive and systematic application of bispectral analysis to the LR04 benthic foraminifera stack. Bispectral analysis allows one to evidence so-called transfers of energy between different frequencies, and may therefore provide support for interpreting non-linear phenomena known to occur in a system of which we can observe time series. Sections 1 and 2 are devoted to context and methodology, and the main results are given in section 3. Section 4 briefly comments on the suitability of the approach, and section 5 suggests possible climate mechanisms.

[Figure]

As pointed out by the authors, this is not the first time that bispectral analysis is being applied to palaeoclimatic time series. Earlier attempts are due to Teresa Hagelberg in the early 1990s and it is nice to see here an up-to-date application of this technique, illustrated by carefully prepared figures (key Figures are 6, 7 and 9). I have, however, a number of comments which I believe pertain to quite fundamental issues, but which nevertheless may be addressed by the authors.

**2 Major Comments**

1. First, the concepts of "energy" and "energy conservation" need to be clarified. In wave theory, the Fourier energy (square of amplitude) is directly interpretable as kinetic energy. The concept of energy conservation therefore has straightforward meaning. In palaeoclimates, the amplitude of a precession beating is not an energy of that form. Therefore, why energy transfers should be conservative is not immediately obvious. If I understood correctly, the specific choice of the weight (p. 7, line 2) enforces conservation, but again the physical justification is unclear. Similarly, the authors follow the state-of-the art literature and focus on the imaginary part of the bispectrum, but as I understood it the physical rationale for focusing on the imaginary part is in fact grounded in wave theory. Why would we focus on the imaginary part in the present context?
Perhaps the reader would be reassured to see the bispectral analysis for typical transformation known to be relevant for palaeoclimate dynamics. What happens with bioturbation (which one might intuitively see as a form of non-conservation, or dissipation)? How does bispectral analysis identify demodulation (precession beating being transformed in a response at the period of the beating). What is happening at a period doubling bifurcation? In other words, we need a user's guide, a reading key of the bispectrum that is well suited to the phenomenology of Pleistocene dynamics. Perhaps these simple examples will also help the reader

understand why the focus should be set on the imaginary part of the bispectrum.

2. Still in relation with the specific phenomenology of palaeoclimate dynamics, it is important to distinguish 'cycle' and 'frequency'. A saw-tooth signal of 100-ka long is the manifestation of one cycle, that is, a succession of events that form a phenomenon (e.g.: the ice-age cycle). Yet the Fourier decomposition of this signal will feature multiple frequencies (an infinite, countable number of them). Hence, a Fourier peak does not necessarily correspond to what we would like to call a 'cycle' or a 'cyclicity' in palaeoclimate dynamics. I am a bit worried about the numerous references to a 28-kyr cycle. Wouldn't it be the main merit of bispectrum analysis to show how frequencies appear in the spectrum and how they are linked to other? In other other words, isn't it precisely the purpose of bispectrum analysis to help one distinguish a frequency from a cycle? (if two frequencies are strongly linked, they are part of a same cycle).

3. It is fine in an exploratory paper to focus on one record, here the LR04 stack. However, the possible pitfalls associated with the way the record for this specific application need be better discussed. The chronology of the LR04 stack was established by tuning the record on the output of a simple ice-age model driven by mid-June insolation (the Imbrie and Imbrie 1980 model), with different time constants for the early and late Pleistocene. By design, this approach tends to concentrate power on astronomical bands, with consequences on the bispectrum which are hard to fully anticipate. On the other hand, the process of stacking different records may have unintended effects on the relative weights between the precession and obliquity components (precession being harder to detect, it may be damaged by a stacking process that favours the visible obliquity signal), and, again, consequences on bispectrum hard to anticipate. Precession signals are also relatively more affected than obliquity's by mixing processes such as biturbation. Hence, I found a bit hasty and not entirely convincing the author's conclusion that stacked records are the best material for their application (p. 20). Splicing

high-resolution, carefully chosen records might in fact be an equally attractive choice.

4. I must confess being quite critical about section 5. The mechanisms for the explanation of the findings are unnecessarily speculative and slightly misinformed, and seem to me to do more harm than good to the credibility of the paper. A word about the "precession motor", first. Clearly precession has various possible effects on ice ages dynamics, via the local insolation forcing, possibly the hydrological cycle, why not the carbon or methane cycles. Hence, focusing on monsoon dynamics is unnecessarily reductive. The simulations presented by Werner et al., 2001 suggest that less than 10 % of the precipitation falling on Greenland on in Eastern Canada is of tropical origin. The article is a bit dated but the order of magnitude must be valid. Hence, monsoon might have a direct effect on ice accumulation balance, but the results presented here provide no argument to see it as a dominant one. Likewise, the reference to a "resonance of crustal sinking" is, again, unnecessarily sophisticated. Physicists and glaciologists working on ice ages broadly agree that terminations are the manifestation of some 'non-linear effect' expressing the instability glacial maxima, and the debate is about the mechanisms of instability (ice-sheet dynamics, ocean and carbon cycle, tectonic $CO_2$ release). Again, the contributions or relative importance of these mechanisms cannot be investigated on the basis of a single record, whatever analysis technique is being used. Finally, the point 5.2.3 about the "climatic and tectonic boundary conditions" is a bit verbose. A quick glance at the LR04 immediately reveals an evolutionary process, which indeed, is being attributed to tectonic changes with perhaps some evolutionary contribution. The authors are citing many references but the context and the purpose of these references is not always clear, and do not relate to an information that bispectrum analysis would have specifically enlightened. In summary, how the bispectrum analysis may contribute to the identification of ice-age dynamics needs to be thought of

better. It seems that the main (and really nice) contribution of bispectrum is to act as a powerful test of dynamical system models of Pleistocene climate dynamics.

1. p. 4 l. 5: follow THE convention

2. equation 2: what is the meaning of $H^3$?

3. p. 6 l. 4: the reference to Fig. 4a is not straightforward. Perhaps say in more plain language what the reader is supposed to look at on the Figure.

4. p. 7 l. 1: "Therefore, we make minimum assumptions and use a coupling coefficient that only corrects for a frequency of $W_(f1, f2) = (f_1 + f_2)$". This seems to be a key passage, of which the implications are not immediately clear to the non-expert. Why does it enforce energy conservation (perhaps this can be explained simply if we consider that the rate of energy loss is counted by cycle), and why having energy conservation allows for "qualitative interpretation". Again, this links with major comment 1. above, the need to explain in simple term what is "energy", and how the imaginary part of the triad interaction is an interesting *qualitative* indicator of energy transfers (comment applies also to p.4 ll. 6-11).

5. p. 8, l. 19: "Nonsinusoidal cycle shapes are generally a good indicator for the successful application of higher order spectral analysis". Ambiguous sentence. If nonsinusoidal cycles are in the record (quite evidently, late Pleistocene cycles are asymmetric), in what sense does it tell us something about the "successful application" of whatever technique?

6. p. 12: "We only document very minimal direct fuelling of eccentricity-paced climate cycles by precession-paced climate cycles in this zone." Can we imagine that this result is influenced by the fact that individual precession cycles are poorly resolved? (bioturbation, undesired effects of stacking).

7. p. 13: The purpose of the reference to Ahn et al., 2017 is not very clear since it seems that the authors have used the original LR04 stack (hence, Lisiecki and Raymo, 2005).

8. p. 13: Section 3.4.2: another confusing point for the non-expert. Given that weights where chosen such that energy is conserved, how could energy not be conserved? A numerical artifact?

9. p. 13, l. 17: "A comparison of conservativities indicates that approximately similar amounts of energy are exchanges in interactions involving obliquity, as in those involving eccentricity". typo: exchanges -> exchanged. The meaning could also be clearer. First, conservativity is a non-standard noun which is not defined in the manuscript (the word appears also in legend of Figure 5). Next, are we speaking of interactions with precession, i.e., are we comparing interactions between precession and obliquity, vs precession and eccentricity? And, again, some more intuitive meaning of "interaction" in the present context (perhaps with a simple example) would be really helpful.

10. p. 15. l. 2: Their may be some confusion between the notion of "reproducibility" (ability to "reproduce" the results based using the data and methodology printed in the manuscript), and "robustness" (insensitivity of results to methodological aspects seemingly unimportant).

11. Figure 1: what are the contours on the continuous wavelet transform plot?

12. Figure 5: the first bit is cryptic: "Input → "black box" climate → output"

Again, the application of bispectrum analysis is promising and interesting and I would definitely encourage the readers to revise the manuscript. Not much revision may be needed in fact. Focus on the methodology, provide a good 'reading key' so that the naive reader understands better the meaning and implication of the notion of 'energy

transfer' in the specific context of palaeoclimate dynamics, and downplay the mechanistic interpretation, which is too speculative and out of scope. Good luck !

**3 References**

- Werner M., M. Heimann and G. Hoffmann (2001), Isotopic composition and origin of polar precipitation in present and glacial climate simulations, Tellus B: Chemical and Physical Meteorology, (53) 53–71 doi:10.3402/tellusb.v53i1.16539

---

## Referee Comment (RC2) · Mathieu Martinez (Referee) · 12 Jul 2019

1. General comments

Drs. Liebrand and de Bakker provided here new, original statistical analyses of the LR04 d18O stacking to document the non linear interactions between the Milankovitch cycles which lead to the generation of new cycles in the palaeoclimatic data and lead to power transfer from the precession band (dominant in the insolation series) to the obliquity and to the 100-kyr eccentricity cycles. In particular, bispectra are used to observe non-linearities between the insolation forcing and the d18O series, which is an excellent and original idea in palaeoclimatology. However, I find section 3 of the manuscript hard to read to someone who is not familiar with the reading the interpre-

tation of bispectra despite I could see the authors made many efforts to make their paper accessible. In section 3.2., I do not understand what is reference to calculate the gains and the losses of energy and I do not understand how the authors can find this information in the bispectra. This step must be clear for any reader to then completely follow the result description in section 3.4.

I can see that bispectra document the energy (or power) transfers and the evolutive spectral analyses seem to document these power transfers very clearly. However, I do not understand how the bispectra contribute in understanding the mechanisms of non linearities in the d18O already evoked in prior publications and this needs to be clarified.

In summary, much clarification is needed to allow a larger community to access this enthusiasming way to observe cycles in sedimentary series and observe their interactions. I thus suggest this manuscript deserves to be published after revisions will be done.

The authors can find more specific comments below:

2. Specific comments

Throughout the manuscript, the term "energy transfer" is used. What do the authors refer to when they use this terminology? This must be more clearly stated, unless I missed it in the manuscript.

In page 3, line 26, the author mention they used SiZer to resample every 1 kyr. What is the method used by SiZer to resample? Is it a linear resampling? Is it another method of resampling? Can the authors write exactly the method of resampling because it can impact the spectrum at high frequencies.

In page 4, line 13, what do the authors call "time averaging operator"?

Section 3.2. ("Bispectra of Pliocene and Pleistocene climate cycles") is hard to follow in my point of view at least for the following reasons:

The authors mention gain or loss of eneregy. Gain or loss should be a difference compared to a reference. What is the reference used to calculate these gains and losses? The authors mention positive or negative interactions, e.g. (page 8, lines 24-25): "negative interactions are concentrated at and between triads along the from ðİŘţ (40âĘŚ, ∞âĘŚ, 40âĘŞ) to ðİŘţ(40âĘŚ, 40âĘŚ, 20âĘŞ)". I do not know where to observe this in Figure 4. Can the authors either explain this with a theoretical example easy to understand prior to the real data or at least show where to observe this in Figure 4?

From these two examples, I think much effort have to be made to guide step-by-step a reader who is not familiar in the use and interpretation of bispectra. Otherwise section 3.4., which describes the results of bispectra, will remain unacessible for many readers. So, I suggest more step by step explanation to make easier the observation and the interpretation of the bispectra.

In Figure 5, I do not understand what the authors refer to by writing "Input –> "black box" climate –> output". What do the authors mean by "black box" here?

Still in Figure 5, I would clarly state what conservative net energy transfer means

In Figures 5 and 6, I would label the freqencies on which energy transfers occur

In section 5.2.1. "Based on the bispectral results, we infer that during the Pliocene and Early Pleistocene this predominantly monsoonally-driven precession motor fuels the 40-kyr obliquity-paced ice age cycles, aided by more linear climatic-cryospheric responses resulting from variability in insolation at this periodicity, especially at higher latitudes" » I do not really understand how the authors can deduce that from bispectral analyses. The authors can of course observe transfers of power from the precession to the obliquity band, but how can they link that to the moisture and heat transfer at low latitude? There is a step I do not understand. Comparatively, the interlatitudinal insolation gradient evoked by Bosmans et al. (2015) appears much more intuitive and easy to understand.

I experience the same feeling with section 5.2.2.: how the power transfer observd in the bispectra can help in linking the transfer from the obliquity to the eccentricity with crustal sinking and delayed rebound? I think the authors need to clarify how the bispectra can contribute to the debate

3. Technical corrections

In Figure S3, labels a, b, c in the caption do not correspond with the panels in the figure. Can the author correct that?

4. References

Bosmans, J.H.C, Hilgen, F.J., Tuenter, E., Lourens, L.J., 2015. Obliquity forcing of low-latitude climate. Clim. Past, 11, 1335-1346.

―――――――――――――――――――――――

---

## Author Comment (AC1) · 11 Aug 2019

**1. Summary**

The authors present an extensive and systematic application of bispectral analysis to the LR04 benthic foraminifera stack. Bispectral analysis allows one to evidence so-called transfers of energy between different frequencies, and may therefore provide support for interpreting nonlinear phenomena known to occur in a system of which we can observe time series. Sections 1 and 2 are devoted to context and methodology, and the main results are given in section 3. Section 4 briefly comments on the suitability of the approach, and section 5 suggests possible climate mechanisms.

As pointed out by the authors, this is not the first time that bispectral analysis is being applied to palaeoclimatic time series. Earlier attempts are due to Teresa Hagelberg in the early 1990s and it is nice to see here an up-to-date application of this technique, illustrated by carefully prepared figures (key Figures are 6, 7, and 9). I have, however, a number of comments which I believe pertain to quite fundamental issues, but which nevertheless may be addressed by the authors.

We find it interesting to learn that R1 considers Figures 6, 7, and 9 as key. We consider Figures 4, 5, and 8 most informative.

**2. Major Comments**

1. First, the concepts of "energy" and "energy conservation" need to be clarified. In wave theory, the Fourier energy (square of amplitude) is directly interpretable as kinetic energy. The concept of energy conservation therefore has straightforward meaning. In palaeoclimates, the amplitude of a precession beating is not an energy of that form.

This is a very fundamental point that R1 raises. Energy and energy conservation for paleoclimatic case studies are dependent on the record that is being analysed. Here, we solely focus on the LR04 benthic foraminiferal oxygen isotope stack and "energy" is given as a function of the variability in benthic foraminiferal $\delta^{18}O$, i.e., expressed in $‰^3 \text{ kyr}^{-2}$ in the bispectrum, and as $‰^3$ when integrating over the bispectrum. However, when other palaeoclimatic time series are considered, the "energy" units will change accordingly.

With respect to the LR04 record: this time series is a globally averaged signal of land-ice volumes and deep-sea temperatures combined. Variability in benthic foraminiferal $\delta^{18}O$ is largely the result of a nonlinear response of the climate-cryosphere system to the changes in

the distribution of Earth's incoming solar radiation (given in the energy units W m$^{-2}$), which is often represented by insolation for a particular latitude (e.g. 65° N).

How energy is transferred from Earth's total energy budget (in W s$^{-1}$, i.e., Joule) into variability in the globally averaged $\delta^{18}O$ record (in ‰ Vienna Pee Dee Belemnite, VPDB), is depending on "…many climatologic and oceanographic, biologic, sedimentologic and lithologic processes…" (see last line of Section 4.2). It is not the purpose of this study to quantify these Earth-internal processes further. We describe "energy" and "energy conservation" merely qualitatively (i.e., translate asymmetry into a loss and a gain at particular frequencies present in the LR04 stack), without scaling them to the power spectrum of the LR04 stack, to the power spectrum of insolation at e.g. 65° N, or to Earth's total energy budget over a given time. These may be objectives for follow-up studies.

> Therefore, why energy transfers should be conservative is not immediately obvious. If I understood correctly, the specific choice of the weight (p. 7, line 2) enforces conservation, but again the physical justification is unclear.

Within the climate system energy losses (i.e., ultimately to space) can occur at numerous stages. These losses will lead to the formation of a particular palaeoclimate record. However, in bispectral analysis and the calculation of the transfer term, these earlier energy losses are not resolved because the bispectrum only considers the available time series, and not what happened beforehand.

When performing bispectral analysis on a specific palaeoclimate record, the conservation of energy during exchanges is assumed (i.e., enforced, obtained, implied) by correcting the energy gains and losses of a particular triad interaction by the values of the frequencies that are involved. This correction is similar/comparable to the Boussinesq scaling used for computing energy exchanges among ocean waves (e.g., (Herbers and Burton, 1997; Herbers et al., 2000)), and allows for qualitative interpretations (i.e., energy gains and losses across frequencies are scaled to one other) (see Section 2.4.1., Fig. 5b, and Fig. 8).

When subsequently making the step to scale these energy exchanges to absolute energy exchanges and make them directly comparable with the time-evolutive gradient of the power spectral density within this specific paleoclimatic record being analysed (i.e., to be able to explain the changes observed within the record through time), the bispectral exchanges have to be corrected for physical processes that play a role in the strength of these exchanges. The dissipation term itself is a completely separate term from the energy conservation that we enforce during triad interactions as documented in the bispectrum (see also Eq. 1 in (Herbers et al., 2000)).

We forego the scaling to absolute transfers here, because of the many unknown/poorly constrained physical, chemical, biological, sedimentological and lithological processes that affect absolute $\delta^{18}O$ values of the globally integrated LR04 record (see Section 4.2). Further research is needed to advance on this point and obtain estimates of the absolute energies that are exchanged.

To further clarify this point, about the assumed energy conservation in nonlinear triad interactions, we have added text ("if we assume a simple coupling coefficient between frequencies") to Section 2.3.1., added "assumed" to Section 3.4.2. and to Section 6., replaced "using" by "assuming" in Section 4.2.

Similarly, the authors follow the state-of-the art literature and focus on the imaginary part of the bispectrum, but as I understood it the physical rationale for focusing on the imaginary part is in fact grounded in wave theory. Why would we focus on the imaginary part in the present context?

The focus on the imaginary part of the bispectrum is not grounded in wave theory, but in bispectral theory. At equal amplitudes, more energy is transferred among frequencies for time series characterized by asymmetric (imaginary part) than for skewed (real part) wave forms/cycle shapes (approx. an order of magnitude difference). Despite this strong focus on the imaginary part of the bispectrum, we do not rule out a (probably much smaller) contributing role for the real part of the bispectrum in describing (even more) energy transfers. This is a potential topic for future research. (See the first point in the Outlook, i.e., Section 6, and Supp. Fig. 1 and 2).

We agree with R1 that the physical rationale for why nearshore waves are asymmetric is much better understood than why climate cycles are asymmetric. See e.g. the comparison of model to flume/beach data presented in de Bakker et al (2016). In this study, for palaeoclimatic interpretations, we speculate that the asymmetry in the LR04 time series is mainly due to nonlinear (positive) ice feedbacks (albedo, inertia of large ice volumes, land-ice mass-loading threshold) that causes a phase-lag with respect to precession and obliquity, and a phase-coupling with respect to ~110-kyr eccentricity.

We have clarified this point in the text by adding "if time series are dominated by asymmetric wave forms/cycle shapes" to the introduction, and by rephrasing the first bullet point of the Outlook (Section 6).

Perhaps the reader would be reassured to see the bispectral analysis for typical transformation known to be relevant for palaeoclimate dynamics. What happens with bioturbation (which one might intuitively see as a form of non-conservation, or dissipation)? How does bispectral analysis identify demodulation (precession beating being transformed in a response at the period of the beating). What is happening at a period doubling bifurcation? In other words, we need a user's guide, a reading key of the bispectrum that is well suited to the phenomenology of Pleistocene dynamics. Perhaps these simple examples will also help the reader understand why the focus should be set on the imaginary part of the bispectrum.

The bispectrum is an accepted method in research fields ranging from nearshore waves, neurology, cardiology, to economics, etc. The extension of these (advanced) techniques to palaeoclimatic problems is one of the latest for bispectral applications in this sequence. It is not the purpose of this study to fully educate the reader in bispectral theory, and some background reading/studying may still be required.

We note that the phenomenology of Pleistocene dynamics is highly proxy record dependant (benthic $\delta^{18}O$ in this case). Hence, there is no single user's guide that will suit all palaeoclimatic purposes. How the bispectrum is precisely affected by the issues raised above (bioturbation, demodulations, period doubling bifurcations), falls outside the scope of this

study. In general, many of these processes will lead to lower signal-to-noise ratios, and hence, more biased results.

We would like to point R1 (and the interested reader) to the "palaeoclimatic" user's guide provided by Hagelberg et al. (1991) and King (1996), who show synthetic examples of frequency and phase (de-) coupled time series and their bicoherence spectra.

2. Still in relation with the specific phenomenology of palaeoclimate dynamics, it is important to distinguish 'cycle' and 'frequency'. A saw-tooth signal of 100-ka long is the manifestation of one cycle, that is, a succession of events that form a phenomenon (e.g.: the ice-age cycle). Yet the Fourier decomposition of this signal will feature multiple frequencies (an infinite, countable number of them). Hence, a Fourier peak does not necessarily correspond to what we would like to call a 'cycle' or a 'cyclicity' in palaeoclimate dynamics. I am a bit worried about the numerous references to a 28-kyr cycle. Wouldn't it be the main merit of bispectrum analysis to show how frequencies appear in the spectrum and how they are linked to other? In other words, isn't it precisely the purpose of bispectrum analysis to help one distinguish a frequency from a cycle? (if two frequencies are strongly linked, they are part of a same cycle).

We agree with R1. However, we choose to use "cycle", "frequency" and "periodicity" more loosely and interchangeably, mainly for textual purposes. We understand that a single frequency (identified in either spectrum or bispectrum) is not necessarily the same as a cycle (identified in a time series), because many cycles are composites of multiple frequencies, most notably skewed, asymmetric, and kurtose cycles.

Despite the (small) differences in the meanings of cycle and frequency, we prefer our less strict semantics to keep the text varied, readable and accessible. However, to also acknowledge the point of R1, we have re-evaluated the usage of "cycle", "frequency" and "periodicity" throughout the manuscript, and in a few instances changed the wording to make specific references to either time (i.e., cyclic phenomena) or frequency domains clearer. We have also added an explanation to Section 2.3.1., clarifying our intended usage of these words.

3. It is fine in an exploratory paper to focus on one record, here the LR04 stack. However, the possible pitfalls associated with the way the record for this specific application need be better discussed. The chronology of the LR04 stack was established by tuning the record on the output of a simple ice-age model driven by mid-June insolation (the Imbrie and Imbrie 1980 model), with different time constants for the early and late Pleistocene. By design, this approach tends to concentrate power on astronomical bands, with consequences on the bispectrum which are hard to fully anticipate. On the other hand, the process of stacking different records may have unintended effects on the relative weights between the precession and obliquity components (precession being harder to detect, it may be damaged by a stacking process that favours the visible obliquity signal), and, again, consequences on bispectrum hard to anticipate. Precession signals are also relatively more affected than obliquity's by mixing processes such as bioturbation. Hence, I found a bit hasty and not entirely convincing the author's conclusion that

stacked records are the best material for their application (p. 20). Splicing high-resolution, carefully chosen records might in fact be an equally attractive choice.

The main rationale to use the LR04 stack in this study is the high signal-to-noise ratio and relatively accurate and precise ages, given the abovementioned tuning assumptions, of which we are fully aware (see caption to Figure 1). We agree with R1 that this does not solely result in benefits, but also in some loss of signal, especially at the higher (precession) frequencies (see also (Huybers and Wunsch, 2004)).

Therefore, we have deleted "Therefore, we argue that for these purposes data stacks are preferred" from Section 6 (last bullet point). In fact, application of bispectra to individual records may prove fruitful for future studies.

4. I must confess being quite critical about section 5. The mechanisms for the explanation of the findings are unnecessarily speculative and slightly misinformed, and seem to me to do more harm than good to the credibility of the paper. A word about the "precession motor", first. Clearly precession has various possible effects on ice ages dynamics, via the local insolation forcing, possibly the hydrological cycle, why not the carbon or methane cycles. Hence, focusing on monsoon dynamics is unnecessarily reductive. The simulations presented by Werner et al., 2001 suggest that less than 10 % of the precipitation falling on Greenland on in Eastern Canada is of tropical origin. The article is a bit dated but the order of magnitude must be valid. Hence, monsoon might have a direct effect on ice accumulation balance, but the results presented here provide no argument to see it as a dominant one.

The suggested modelling paper by Werner et al, (2001) is mainly concerned with Greenlandic land-ice isotope composition, and does not seem too relevant to our study. During the current interglacial (and those of the past million years or so), Greenland is still largely glaciated. Hence, the moisture source for Greenland is not very relevant in explaining large land-ice volume fluctuations of the past million year (de Boer et al., 2012). The largest land-ice volumes during Middle and Late Pleistocene glacial maxima were located on the North American and Eurasian continents. Hence, moisture sources for these regions during glacial inceptions and maxima may well be largely temperate to (sub-) tropical in origin. Further evidence for the strength of the precession motor comes from lower latitudes (e.g. the sapropels in the Mediterranean), which show the latitudinal migrations of atmospheric (and oceanic) fronts and associated hydroclimate on these shorter, precession time scales (Bosmans et al., 2015).

However, to acknowledge the uncertainty that remains in the understanding of precipitation sources for large land-ice volumes, we have added a reference to Werner et al, (2001), in addition to references to other, more recent modelling studies.

Likewise, the reference to a "resonance of crustal sinking" is, again, unnecessarily sophisticated. Physicists and glaciologists working on ice ages broadly agree that terminations are the manifestation of some 'non-linear effect' expressing the instability glacial maxima, and the debate is about the mechanisms of instability (ice-sheet dynamics, ocean and carbon cycle, tectonic CO2 release). Again, the

contributions or relative importance of these mechanisms cannot be investigated on the basis of a single record, whatever analysis technique is being used.

We agree with R1 that our analysis does not point to crustal sinking as the mechanism. However, we merely state that the bispectral results obtained in this study are in agreement with a nonlinear mechanism, such as crustal sinking and resonance with eccentricity modulated precession (e.g. following (Pisias et al., 1990; Abe-Ouchi et al., 2013)).

Finally, the point 5.2.3 about the "climatic and tectonic boundary conditions" is a bit verbose. A quick glance at the LR04 immediately reveals an evolutionary process, which indeed, is being attributed to tectonic changes with perhaps some evolutionary contribution. The authors are citing many references but the context and the purpose of these references is not always clear, and do not relate to an information that bispectrum analysis would have specifically enlightened.

For contextual purposes, we thought to briefly (one paragraph only) address the long-term climatic evolution during the Pliocene and Pleistocene, mainly to set clear boundaries on what the bispectrum can–and what it cannot–help to understand better. Especially the comparison of the LR04 spectrum to the LR04 bispectrum is of relevance. The time-evolutive spectral analysis show long term frequency evolutions that are absent in the time-evolutive bispectral analysis (compare Fig. 5b to 5c), suggesting shifts in the response of the climate system that are unrelated to the nonlinear processes described by the bispectrum.

In summary, how the bispectrum analysis may contribute to the identification of ice-age dynamics needs to be thought of better. It seems that the main (and really nice) contribution of bispectrum is to act as a powerful test of dynamical system models of Pleistocene climate dynamics.

We agree with R1 that the bispectrum may serve as a powerful test of dynamical system models (GCMs or conceptual) and help the understanding of the Pliocene and Pleistocene climate system. However, we disagree with R1, and do not find it "unnecessarily speculative" and "misinformed" (see R1s point 4) to then also interpret the bispectral results in terms of dynamics (i.e., mechanisms), despite the fact that these interpretations are speculative.

Given the most thorough higher order spectral description of nonlinearities during the Pliocene and Pleistocene to date, which we present, some speculation on the mechanisms is in place. In fact, we would perceive it as a missed opportunity not to at least attempt to (speculatively) link these new observations and descriptions of nonlinearities to mechanisms that have been proposed in the literature. Throughout the Discussion, we have made it very clear that these interpretations are speculative at best.

1. p. 4 l. 5: follow THE convention

Corrected.

2. equation 2: what is the meaning of $H^3$?

This is indeed a mistake. We have changed $As(x) = \frac{\langle H^3(x-\bar{x})\rangle}{\langle(x-\bar{x})^2\rangle^{3/2}}$ into $As(x) = \frac{\langle H(x-\bar{x})^3\rangle}{\langle(x-\bar{x})^2\rangle^{3/2}}$. This typo was unfortunately not noticed and corrected in in the text of Liebrand et al. (2017).We checked, and the computations that were performed in MATLAB use the latter (correct) formula. $H$ stands for the Hilbert transform.

3. p. 6 l. 4: the reference to Fig. 4a is not straightforward. Perhaps say in more plain language what the reader is supposed to look at on the Figure.

Reading bispectra is not straightforward indeed, which is why we focus the interpretations of the bispectra on the integrations, which transpose the frequency-frequency domain into the time-frequency domain. We did want to show a few clear examples of bispectra, to familiarize the reader with the analysis underpinning the results that are presented later on in the manuscript.

To aid the understanding of Figure 4, we have restructured Section 2.3.2., and moved last paragraph upward. By reordering this section, we now first explain how to read sum frequencies. This should make the reference to Figure 4a also more accessible. All frequencies and periodicities were already given in bispectral notation, which point the reader to the correct "blue area" in the bispectrum.

4. p. 7 l. 1: "Therefore, we make minimum assumptions and use a coupling coefficient that only corrects for a frequency of W(f 1, f 2) = (f1 + f2)". This seems to be a key passage, of which the implications are not immediately clear to the non-expert. Why does it enforce energy conservation (perhaps this can be explained simply if we consider that the rate of energy loss is counted by cycle), and why having energy conservation allows for "qualitative interpretation"? Again, this links with major comment 1. above, the need to explain in simple term what is "energy", and how the imaginary part of the triad interaction is an interesting qualitative indicator of energy transfers (comment applies also to p.4 l. 6-11).

See our rebuttal to R1's Main Point 1 above.

5. p. 8, l. 19: "Nonsinusoidal cycle shapes are generally a good indicator for the successful application of higher order spectral analysis". Ambiguous sentence. If nonsinusoidal cycles are in the record (quite evidently, late Pleistocene cycles are asymmetric), in what sense does it tell us something about the "successful application" of whatever technique?

Higher order spectra describe nonsinusoidality. However, we agree with R1 that nonsinusoidality on its own, is not sufficient for the successful application of higher order spectra.

We have rephrased this sentence to remove the ambiguity.

6. p. 12: "We only document very minimal direct fuelling of eccentricity-paced climate cycles by precession-paced climate cycles in this zone." Can we imagine that

this result is influenced by the fact that individual precession cycles are poorly resolved? (bioturbation, undesired effects of stacking).

This may well be the case. The LR04 stack is indeed a globally integrated, land-ice volume dominated record, which may have attenuated precession variability compared to other proxy records, and especially compared to insolation variability at any particular latitude. However, despite the likely bias of this proxy to variability at the lower frequencies, we do see energy transfers from precession to obliquity periodicities (e.g. see Zone 5, OOP), and of obliquity to eccentricity periodicities (e.g. see Zone 2, EEO). Therefore, we argue, that the lack of direct "fuelling" of eccentricity variability by precession (see Zone 6, PEP) is a valid observation,

In the Results Chapter, we prefer to observe and describe without too much interpretation, and have therefore left the text as is. In the Outlook (Section 6) we argue that further higher order spectral analyses, on climate time series that are less land-ice volume dominated, may well be insightful.

7. p. 13: The purpose of the reference to Ahn et al., 2017 is not very clear since it seems that the authors have used the original LR04 stack (hence, Lisiecki and Raymo, 2005).

We have rephrased this sentence by replacing "are" with "may be".

8. p. 13: Section 3.4.2: another confusing point for the non-expert. Given that weights where chosen such that energy is conserved, how could energy not be conserved? A numerical artefact?

See our rebuttal to R1's Main Point 1 above.

9. p. 13, l. 17: "A comparison of conservativities indicates that approximately similar amounts of energy are exchanges in interactions involving obliquity, as in those involving eccentricity". typo: exchanges -> exchanged.

Corrected

The meaning could also be clearer. First, conservativity is a non-standard noun which is not defined in the manuscript (the word appears also in legend of Figure 5).

We have now defined "conservativity" (Section 3.4.2.) and rephrased the figure captions of Fig. 9 and Fig. 10 (N.B. not Fig. 5).

Next, are we speaking of interactions with precession, i.e., are we comparing interactions between precession and obliquity, vs precession and eccentricity?

This sentence (starting with "A comparison of conservativities…") refers to Figure 10d, in which we compare conservativities of the recombined zones, that contain at least one

precession, obliquity, or eccentricity component (See the first sentence of Section 3.4.3). The answer to R1's question is no.

We have rephrased the text to make this point clearer.

> And, again, some more intuitive meaning of "interaction" in the present context (perhaps with a simple example) would be really helpful.

We added a definition of "triad interaction" to Section 2.3.1. However, Figure 10 shows the recombined zonal integrations over the imaginary part of the bispectrum. Frequencies participating in multiple triad interactions are summed and may therefore no longer be visible.

We have added "(triad)" to this particular sentence, to remind the reader of the link to the bispectrum that underpins these computations of energy exchanges.

> 10. p. 15. l. 2: There may be some confusion between the notion of "reproducibility" (ability to "reproduce" the results based using the data and methodology printed in the manuscript), and "robustness" (insensitivity of results to methodological aspects seemingly unimportant).

Throughout the text we have replaced "reproduce" with "robust/robustness".

> 11. Figure 1: what are the contours on the continuous wavelet transform plot?

The black contours represent 95% significance. We have added this information to the figure caption.

> 12. Figure 5: the first bit is cryptic: "Input → "black box" climate → output"

To clarify this figure caption, we have added the relevant panel call-outs ((a), (b), (c)). The meaning of the figure of well-explained in the rest of the caption.

We are aware that strictly speaking bispectra are an "output" analysis, however, our framing here, as a window into the "black box" response, corresponds to the framing of the paper; namely that bispectra 'show how' ice ages are fuelled.

Again, the application of bispectrum analysis is promising and interesting and I would definitely encourage the readers to revise the manuscript. Not much revision may be needed in fact. Focus on the methodology, provide a good 'reading key' so that the naive reader understands better the meaning and implication of the notion of 'energy transfer' in the specific context of palaeoclimate dynamics, and downplay the mechanistic interpretation, which is too speculative and out of scope. Good luck!

Energy and energy transfer do not have a specific context that relates to "palaeoclimate dynamics". These are bispectral properties specific to each proxy time series. See also our rebuttal to R1's Main Point 1 above.

**3. References**

Werner M., M. Heimann and G. Hoffmann (2001), Isotopic composition and origin of polar precipitation in present and glacial climate simulations, Tellus B: Chemical and Physical Meteorology, (53) 53–71 doi:10.3402/tellusb.v53i1.16539

We have included this reference in the manuscript.

**Rebuttal references:**

Abe-Ouchi, A., Saito, F., Kawamura, K., Raymo, M. E., Okuno, J., Takahashi, K., and Blatter, H.: Insolation-driven 100,000-year glacial cycles and hysteresis of ice-sheet volume, Nature, 500, 190–194, https://doi.org/10.1038/nature12374, 2013.

Bosmans, J. H. C., Drijfhout, S. S., Tuenter, E., Hilgen, F. J., and Lourens, L. J.: Response of the North African summer monsoon to precession and obliquity forcings in the EC-Earth GCM, Clim Dynam, 44, 279–297, https://doi.org/10.1007/s00382-014-2260-z, 2015.

de Bakker, A. T. M., Tissier, M. F. S., and Ruessink, B. G.: Beach steepness effects on nonlinear infragravity-wave interactions: A numerical study, J Geophys Res-Oceans, 121, 554–570, https://doi.org/10.1002/2015jc011268, 2016.

de Boer, B., van de Wal, R. S. W., Lourens, L. J., and Bintanja, R.: Transient nature of the Earth's climate and the implications for the interpretation of benthic delta O-18 records, Palaeogeogr Palaeocl, 335, 4–11, 10.1016/j.palaeo.2011.02.001, 2012.

Hagelberg, T., Pisias, N., and Elgar, S.: Linear and nonlinear couplings between orbital forcing and the marine $^{18}$O record during the late Neogene, Paleoceanography, 6, 729–746, https://doi.org/10.1029/91PA02281, 1991.

Herbers, T. H. C., and Burton, M. C.: Nonlinear shoaling of directionally spread waves on a beach, J Geophys Res-Oceans, 102, 21101–21114, https://doi.org/10.1029/97jc01581, 1997.

Herbers, T. H. C., Russnogle, N. R., and Elgar, S.: Spectral energy balance of breaking waves within the surf zone, Journal of Physical Oceanography, 30, 2723–2737, https://doi.org/10.1175/1520-0485(2000)030<2723:SEBOBW>2.0.CO;2, 2000.

Huybers, P., and Wunsch, C.: A depth-derived Pleistocene age model: Uncertainty estimates, sedimentation variability, and nonlinear climate change, Paleoceanography, 19, https://www.doi.org/10.1029/2002pa000857, 2004.

King, T.: Quantifying nonlinearity and geometry in time series of climate, Quaternary Science Reviews, 15, 247–266, https://doi.org/10.1016/0277-3791(95)00060-7, 1996.

Liebrand, D., de Bakker, A. T. M., Beddow, H. M., Wilson, P. A., Bohaty, S. M., Ruessink, G., Pälike, H., Batenburg, S. J., Hilgen, F. J., Hodell, D. A., Huck, C. E., Kroon, D., Raffi, I., Saes, M. J. M., van Dijk, A. E., and Lourens, L. J.: Evolution of the early Antarctic ice ages, Proceedings of the National Academy of Sciences of the United States of America, 114, 3867–3872, https://doi.org/10.1073/pnas.1615440114, 2017.

Pisias, N. G., Mix, A. C., and Zahn, R.: Nonlinear response in the global climate system: evidence from benthic oxygen isotopic record in Core Rc13-110, Paleoceanography, 5, 147–160, https://doi.org/10.1029/PA005i002p00147, 1990.

Werner, M., Heimann, M., and Hoffmann, G.: Isotopic composition and origin of polar precipitation in present and glacial climate simulations, Tellus B, 53, 53–71, https://doi.org/10.3402/tellusb.v53i1.16539, 2001.

We would like to take this opportunity to thank M. Crucifix for his constructive feedback.

---

## Author Comment (AC2) · 11 Aug 2019

**1. General comments**

Drs. Liebrand and de Bakker provided here new, original statistical analyses of the LR04 $\delta^{18}$O stacking to document the nonlinear interactions between the Milankovitch cycles which lead to the generation of new cycles in the palaeoclimatic data and lead to power transfer from the precession band (dominant in the insolation series) to the obliquity and to the 100-kyr eccentricity cycles. In particular, bispectra are used to observe nonlinearities between the insolation forcing and the $\delta^{18}$O series, which is an excellent and original idea in palaeoclimatology.

In addition to R2's summary, we would like to emphasize that we apply bispectral analysis only to the LR04 record, and thereby describe nonlinear interactions among climate frequencies as present in this data set (i.e., a transform of the asymmetric cycle geometry), and not between insolation and $\delta^{18}$O. The comparison to insolation forcing is merely qualitative (see Fig. 6), and is presented for comparison based on a theoretical/physical understanding of astronomical climate forcing.

However, I find section 3 of the manuscript hard to read to someone who is not familiar with the reading the interpretation of bispectra despite I could see the authors made many efforts to make their paper accessible.

To clarify the interpretation of bispectra we provide a thoughtfully constructed reading key in Section 2.3.2. ("Interpreting the bispectrum"). In this section we also refer to Figure 4, to support the interpretation of the bispectrum. This information is crucial for understanding the results (Section 3) and the interpretation of the bispectra.

In section 3.2., I do not understand what is reference to calculate the gains and the losses of energy and I do not understand how the authors can find this information in the bispectra. This step must be clear for any reader to then completely follow the result description in section 3.4.

To read Section 3.2., and understand how gains and losses are computed we refer R2 (and the interested reader) to Section 2.3.2., in which we explain in more plain language how the bispectrum is interpreted, and how the bispectral notation should be read.

In short: an energy gain (EG) is depicted by the warm colours (red), whereas an energy loss (EL) is represented by cold colours (blue). An energy gain at one frequency results in energy

losses at two other frequencies and vice versa: $EG_{f1}+EG_{f2} = EL_{f3}$ and $EL_{f1}+EL_{f2} = EG_{f3}$. The "energy" reference (i.e., amount of energy that is exchanged in such a nonlinear triad interaction) is given as a function of the variability in benthic foraminiferal $\delta^{18}O$, expressed in $‰^3 \, kyr^{-2}$ in the bispectrum, and as $‰^3$ when integrating over the bispectrum (as is described in Section 3.4.).

We have added information about these energy units to Section 2.3.1. and Section 2.4.

I can see that bispectra document the energy (or power) transfers and the evolutive spectral analyses seem to document these power transfers very clearly. However, I do not understand how the bispectra contribute in understanding the mechanisms of nonlinearities in the $\delta^{18}O$ already evoked in prior publications and this needs to be clarified.

To date, no similarly detailed (i.e., in a time-evolutive manner) description of nonlinear energy transfers among Pliocene and Pleistocene climate cycles is available. The detailed documentation of these transfers presented in this study 'show how' (i.e., through which cycle-cycle interactions) energy is transferred from the insolation frequencies (mainly precession) to those of the ice ages (40, 80, 120, and 95 kyr), and how these transfers evolve through time. In light of these new results, we deem it valuable to tentatively link them to mechanisms that have previously been proposed in the literature. In our opinion, it would be a missed opportunity not to (at least) attempt to draw further conclusions about potential nonlinear mechanisms, given the best available description of nonlinear energy transfers.

A suggestion for a follow-up study would be to test with fully coupled climate-ice sheet models which mechanisms correspond to the specific frequency interactions we observe in the LR04 stack. We have added this suggestion to the Outlook (see the third bullet point in Section 6).

In summary, much clarification is needed to allow a larger community to access this enthusing way to observe cycles in sedimentary series and observe their interactions. I thus suggest this manuscript deserves to be published after revisions will be done.

On this particular point (i.e., "much clarification"), we disagree with R2. Section 2.3. is concerned with the interpretation of bispectra and was designed with great care. It provides an explanation of the bispectrum, and how to read one. This explanation is more detailed than most existing papers on bispectra. Section 2.3. is written with the specific aim of explaining the bispectrum to the nonexpert, and we believe that after reading this section, the remainder of the text is accessible to most palaeoclimatologists/-oceanographers.

The authors can find more specific comments below:

**2. Specific comments**

Throughout the manuscript, the term "energy transfer" is used. What do the authors refer to when they use this terminology? This must be more clearly stated, unless I missed it in the manuscript.

To further clarify what is meant by "energy transfers" in this particular case study on the LR04 record, we have added a sentence to Section 2.3.1. and to Section 2.4.1.

In page 3, line 26, the author mention they used SiZer to resample every 1 kyr. What is the method used by SiZer to resample? Is it a linear resampling? Is it another method of resampling? Can the authors write exactly the method of resampling? Because it can impact the spectrum at high frequencies.

The SiZer method computes the statistically significant zero crossings of the first and second derivatives of an unevenly sampled time series, to compute a 'family of smooths' that fit these criteria. We used the raw data of the LR04 stack to compute these smooths and selected the smooth (out of 41 smooths) that preserved the most structure in the data, given the 1 kyr resampling resolution. SiZer smooths are not linear interpolations. We refer to Chaudhuri and Marron (1999) for a more detailed description of the SiZer method.

We added information about our smooth selection criteria to Section 2.1.

With respect to the impact on higher frequencies: The highest frequency considered in this study is 100 Myr$^{-1}$, equivalent to a periodicity of 10 kyr. A resampling resolution of 1 kyr (1000 Myr$^{-1}$) yields a Nyquist frequency of 500 Myr$^{-1}$, which is well above that of the cycle frequencies considered in this study (i.e., 0 to 100 Myr$^{-1}$). Furthermore, in an earlier stage of this study, we have performed sensitivity tests for different resampling resolutions (not included in the current study or supplements), and found no difference on the astronomical frequencies considered here.

In page 4, line 13, what do the authors call "time averaging operator"?

This is the window length considered. In this study, most often 668 kyr long windows were used (apart from in the supplements where 500 kyr and 1000 kyr long window lengths are considered).

We have added "(i.e., window length)" to the sentence.

Section 3.2. ("Bispectra of Pliocene and Pleistocene climate cycles") is hard to follow in my point of view at least for the following reasons:

The authors mention gain or loss of energy. Gain or loss should be a difference compared to a reference. What is the reference used to calculate these gains and losses?

See our rebuttal to previous comments by R2 (i.e., the third "General Comment" and first "Specific Comment")

The authors mention positive or negative interactions, e.g. (page 8, lines 24-25): " negative interactions are concentrated at and between triads along the lines from $B_p^{Im}(40\uparrow, \infty\uparrow, 40\downarrow)$ to $B_p^{Im}(40\uparrow, 40\uparrow, 20\downarrow)$". I do not know where to observe this in Figure 4. Can the authors either explain this with a theoretical example easy to understand prior to the real data or at least show where to observe this in Figure 4?

In section 2.3.2. we explain the bispectral notation. This information explains how the bispectrum is read and interpreted. In this section we also give a simple example, not theoretical, but based on Figure 4a.

From these two examples, I think much effort have to be made to guide step-by-step a reader who is not familiar in the use and interpretation of bispectra. Otherwise section 3.4., which describes the results of bispectra, will remain inaccessible for many readers. So, I suggest more step by step explanation to make easier the observation and the interpretation of the bispectra.

See previous rebuttal comments to R2. (i.e., Section 2.3 is key in understanding Section 3).

In Figure 5, I do not understand what the authors refer to by writing "Input –> "black box" climate –> output". What do the authors mean by "black box" here?

We agree with R2 that the title of this figure caption is a bit cryptic. However, this was done on purpose, with the aim to provoke thought about the workings of the Earth System. "Black box" is a commonly used metaphor for a system of which the inner workings are only partially understood. Earth's climate system is such a "black box". Its past behaviour can only be approximated (by proxy records) or understood theoretically/through modelling. Spectra of insolation represent the climate driver (i.e., input) and the spectra of the benthic $\delta^{18}O$ record constitute the "black box" response (i.e., output).

We have now labelled the corresponding panels in the Figure caption.

Still in Figure 5, I would clearly state what conservative net energy transfer means

We have added text in between brackets for the explanation of panel (b).

In Figures 5 and 6, I would label the frequencies on which energy transfers occur

Both frequencies and periodicities are labelled along the axes. Adding numbers within the figures would, in our opinion, make them more cluttered and less easy to read.

In section 5.2.1. "Based on the bispectral results, we infer that during the Pliocene and Early Pleistocene this predominantly monsoonally-driven precession motor fuels the 40-kyr obliquity-paced ice age cycles, aided by more linear climatic-cryospheric responses resulting from variability in insolation at this periodicity, especially at higher latitudes" » I do not really understand how the authors can deduce that from bispectral analyses. The authors can of course observe transfers of power from the precession to the obliquity band, but how can they link that to the moisture and heat transfer at low latitude? There is a step I do not understand. Comparatively, the interlatitudinal insolation gradient evoked by Bosmans et al. (2015) appears much more intuitive and easier to understand.

The modelling study by Bosmans et al. (2015) is concerned with explaining obliquity signals at low latitudes, and suggests that these signals may originate in the (sub-) tropics. However, we aim to explain the transfer of energy from precession to obliquity cycles in a high latitude land-ice volume dominated climate record. Although obliquity is observed at low latitudes, many of the (sub-) tropical palaeoclimate records remain precession dominated (e.g., monsoonal/loess records, sapropels, caves records, etc.). Therefore, we speculate that insolation changes at the lower latitudes (mainly precession paced) may fuel the transport of heat and moisture to the poles, and the build-up on obliquity time scales (with associated energy transfers from precession to obliquity as documented in the bispectra) of Northern Hemisphere land ice. The study of Bosmans (2015) does not include a dynamic ice sheet and can therefore not capture these hypothesised energy transfers.

I experience the same feeling with section 5.2.2.: how the power transfer observed in the bispectra can help in linking the transfer from the obliquity to the eccentricity with crustal sinking and delayed rebound? I think the authors need to clarify how the bispectra can contribute to the debate

We copy our reply to R1, who also raised this point: Our analysis does not point to crustal sinking as the mechanism. However, we merely state that the bispectral results obtained in this study, are in agreement with a nonlinear mechanism, such as crustal sinking and resonance with eccentricity modulation precession (e.g. following (Pisias et al., 1990; Abe-Ouchi et al., 2013)).

**3. Technical corrections**

In Figure S3, labels a, b, c in the caption do not correspond with the panels in the figure. Can the author correct that?

We have corrected the figure caption.

**4. References**

Bosmans, J.H.C, Hilgen, F.J., Tuenter, E., Lourens, L.J., 2015. Obliquity forcing of low-latitude climate. Clim. Past, 11, 1335-1346.

We have added this reference to the manuscript.

**Rebuttal references:**

Abe-Ouchi, A., Saito, F., Kawamura, K., Raymo, M. E., Okuno, J., Takahashi, K., and Blatter, H.: Insolation-driven 100,000-year glacial cycles and hysteresis of ice-sheet volume, Nature, 500, 190–194, https://doi.org/10.1038/nature12374, 2013.

Bosmans, J. H. C., Hilgen, F. J., Tuenter, E., and Lourens, L. J.: Obliquity forcing of low-latitude climate, Clim Past, 11, 1335–1346, https://doi.org/10.5194/cp-11-1335-2015, 2015.

Chaudhuri, P., and Marron, J. S.: SiZer for exploration of structures in curves, J Am Stat Assoc, 94, 807–823, https://doi.org/10.2307/2669996, 1999.

Pisias, N. G., Mix, A. C., and Zahn, R.: Nonlinear response in the global climate system: evidence from benthic oxygen isotopic record in Core Rc13-110, Paleoceanography, 5, 147–160, https://doi.org/10.1029/PA005i002p00147, 1990.

We would like to take this opportunity to thank M. Martinez for his constructive feedback.

---

## Referee Report (RR1)

Review of Liebrand and de Bakker: Bispectra of climate cycles show how ice ages are fuelled

In their revised manuscript, Drs. Liebrand and de Bakker made substantial modifications which clarified the method and the interpretations in terms of climate dynamics. Clarification is still possible by defining terms when they are used for the first time in the manuscript. Nonetheless, the manuscript is a great contribution tothe understandng of Plio-Pleistocene climatic changes and will be suitable for publication after minor revisions would be done. Below are my detailed comments:

1) A possible mistake I saw in the revised version, is the definition of the kurtosis. Kurtosis, which, in the absence of outlier, measures how flat or peaked is the top of a distribution of values, ranges between 1 and the infinite. For a normal distribution, the kurtosis value is 3. For a flat-top distribution, the kurtosis is 1. Then, themore peaked is a distribution, the higher is the kurtosis. Subtracting 3 to the kurtosis value means the normal distribution is centred to 0. In that case, what is measured is called the excess kurtosis which ranges from -2 to infinite. See for instance Collis et al. (1998). If the authors decide to keep this formulation, they should replace "kurtosis" by "excess kurtosis" throughout the text and in Figure 2.

2) **In Fig. 4c and in section 3.2.**, how there can be in the same bispectrum the following triad interactions: $B^{Im}_{p}(40\uparrow, \infty\uparrow, 40\downarrow)$ (Line 20) and $B^{Im}_{p}(40\downarrow, \infty\downarrow, 40\uparrow)$ (Line 24)? ow can a triad interaction be positive and negative in the same time?

3) **Page 7, lines 24:** section 4.3. does not exist. Do the authors refer to secton 3.4. instead?

4) In the method section, I think that defining terms which are not reorganizing ideas or paragraphs would make the text muh straightfoward to read. For instance, „conservative energy exchange" is an important concept in the manuscript. This is mentioned for the time in page 4 but only defined in page 13. In my opinion, the concepts the authors use should be defined when they are mentioned for the time in the manuscript. Below, I list possible rephrasing and additional explanations in the method section. These additional explanation may be more inclusive for readers who are interested in Pleistocene climate changes but feel not confident in higher order statistics. The authors can feel free to account them or modify them if misunderstanding appeared to come from me:

2.2. Quantifying geometries using central moments

The geometry of a distribution of values can be quantified by a infinity of statistical moments. Generally, only the first four moments are used. These four statistical moments are respectively the average, the variance, the skewness (or asymmetry) and the kurtosis. Here, the cycles are assimilated to a distribution of values through time. The skewness is here defined as a disymmetry of a cycle relatively to a horizontal axis (Fig. 2b). The asymetry is here the dissymetry of a cycle through time (Fig. 2c). The kurtosis quantifies the flatness of the extrema of a cycle. Flat-top (flat-bottom) cycles have low-kurtosis values, while sharp-top (sharp-bottom) cycles have a high kurtosis value. Kurtosis ranges from 1 to infinite. A Gaussian curve has a kurtosis value of 3. Thus, the deviation of a curve to the Gaussian shape can be calculate by the "excess kurtosis" which is defined as follows: excess kurtosis = kurtosis − 3. Using third-moment quantities, skewness is determined by Eq. (1)...

**2.3.1. The bispectrum**

The Fourier Transform calculates a spectrum showing the distribution of variance (being related to power and energy, as power is the energy per time unit) with frequency. The Fourier Transforms is however unable to document higher order statistical moments of the signal considered and, for this, higher order spectral analyses must be done. Bispectral analyses describe the distribution of nonsinusoidality with frequency, in both the real and imaginary parts (King, 1996). The skewness of a cycle geometry is related to the real part of the bispectrum while the asymetry is related to its imaginary part (Fig. 2). The bispectrum shows nonconservative, relative energy exchanges among frequencies of a single time series. Conservative energy exchanges are here defined as exchanges of energy and relative energy exchanges as … (*I think this is important to introduce the terms here otherwise, many readers may be lost depending on their own usage of the terminology*). Energy transfers can occur ...

**2.3.2. Interpreting the bispectrum**

**Page 5, Line 30:** „The former are the so-called difference frequencies, while the latter is referred to as a sum frequency": using „former" and „latter" can be confusing beause it is not always to what they refer. I usually tend to avoid these terms. I would write instead: $f1$ and $f2$ are so-called difference frequencies, while $f3$ is referred to as sum frequency".

I find the subsection „2.3.2 Interpreting the bispectrum" very practical and useful. Nonetheless, in this subsection, the notions of energy gains and losses are mentioned, while they are defined further in subsection 2.3.4. In my opinion. It would be more logical to define the bispectrum, the calculation of geometries from the bispectrum, the energy exchanges, and then show how to interpret the bispectrum. Again, this organisation should ensure that terms and concepts are defined when they are mentioned for the first time in the manuscript, which is important.

**Reference:**

Collis, W.B., White, P.R., Hammon, J.K., 1998. Higher-order spectra: the bispectrum and the trispectrum. Mechanical Systems and Signal Processing 12(3), 375-394.

---

## Author Response (AR2)

**Rebuttal to *Second review of* "Bispectra of climate cycles show how ice ages are fuelled" by Diederik Liebrand and Anouk T. M. de Bakker**

**Mathieu Martinez (Referee) R2**

mathieu.martinez@univ-rennes1.fr

Received: 29 September 2019

In their revised manuscript, Drs. Liebrand and de Bakker made substantial modifications which clarified the method and the interpretations in terms of climate dynamics. Clarification is still possible by defining terms when they are used for the first time in the manuscript. Nonetheless, the manuscript is a great contribution to the understanding of Plio-Pleistocene climatic changes and will be suitable for publication after minor revisions would be done. Below are my detailed comments:

We agree with R2 that some further clarifications could be made. Especially with regard to defining terms when they are introduced. We have made changes to the text (see below).

1) A possible mistake I saw in the revised version, is the definition of the kurtosis. Kurtosis, which, in the absence of outlier, measures how flat or peaked is the top of a distribution of values, ranges between 1 and the infinite. For a normal distribution, the kurtosis value is 3. For a flat-top distribution, the kurtosis is 1. Then, the more peaked is a distribution, the higher is the kurtosis. Subtracting 3 to the kurtosis value means the normal distribution is centred to 0. In that case, what is measured is called the excess kurtosis which ranges from -2 to infinite. See for instance Collis et al. (1998). If the authors decide to keep this formulation, they should replace "kurtosis" by "excess kurtosis" throughout the text and in Figure 2.

We agree with R2 that Equation 3 computes excess kurtosis, and we have added this information to the text and Fig. 2.

2) **In Fig. 4c and in section 3.2.**, how there can be in the same bispectrum the following triad interactions: $B^{lm}_p(40\uparrow, \infty\uparrow, 40\downarrow)$ (Line 20) and $B^{lm}_p(40\downarrow, \infty\downarrow, 40\uparrow)$ (Line 24)? How can a triad interaction be positive and negative in the same time?

Good observation of R2. We agree that this is impossible. In fact, this interaction is negative (i.e., blue colours), hence $B^{lm}_p(40\uparrow, \infty\uparrow, 40\downarrow)$ is the correct notation. Where previously, we marked this interaction as positive, we have now corrected the periodicities to the correct positive interaction at $B^{lm}_p(44\downarrow, 405\downarrow, 40\uparrow)$ in the text. Figure 4c remains the same.

3) **Page 7, lines 24:** section 4.3. does not exist. Do the authors refer to section 3.4. instead?

We agree with R2 that 4.3 does not exist. We meant 4.2 and have updated the text.

4) In the method section, I think that defining terms which are not reorganizing ideas or paragraphs would make the text much straightforward to read. For instance, "conservative energy exchange" is an important concept in the manuscript. This is mentioned for the time in page 4 but only defined in   page 13. In my opinion, the concepts the authors use should be defined when they are mentioned for  the time in the manuscript. Below, I list possible rephrasing and additional explanations in the   method section. These additional explanations may be more inclusive for readers who are interested  in Pleistocene climate changes but feel not confident in higher order statistics. The authors can feel   free to account them or modify them if misunderstanding appeared to come from me:

We agree with R2 that new terms/concepts can best be explained when introduced. (see below)

2.2. Quantifying geometries using central moments

The geometry of a distribution of values can be quantified by an infinity of statistical moments. Generally, only the first four moments are used. These four statistical moments are respectively the average, the variance, the skewness (or asymmetry) and the kurtosis. Here, the cycles are assimilated to a distribution of values through time. The skewness is here defined as a dissymmetry of a cycle relatively to a horizontal axis (Fig. 2b). The asymmetry is here the dissymmetry of a cycle  through time (Fig. 2c). The kurtosis quantifies the flatness of the extrema of a cycle. Flat-top (flat-   bottom) cycles have low-kurtosis values, while sharp-top (sharp-bottom) cycles have a high kurtosis  value. Kurtosis ranges from 1 to infinite. A Gaussian curve has a kurtosis value of 3. Thus, the  deviation of a curve to the Gaussian shape can be calculate by the "excess kurtosis" which is  defined as follows: excess kurtosis = kurtosis – 3. Using third-moment quantities, skewness is   determined by Eq. (1) ...

We appreciate this suggestion and we have partly incorporated this text into the Method section.

2.3.1. The bispectrum

The Fourier Transform calculates a spectrum showing the distribution of variance (being related to power and energy, as power is the energy per time unit) with frequency. The Fourier Transforms is however unable to document higher order statistical moments of the signal considered and, for this, higher order spectral analyses must be done. Bispectral analyses describe the distribution of nonsinusoidality with frequency, in both the real and imaginary parts (King, 1996). The skewness of a cycle geometry is related to the real part of the bispectrum while the asymmetry is related to its imaginary part (Fig. 2). The bispectrum shows nonconservative, relative energy exchanges among frequencies of a single time series. Conservative energy exchanges are here defined as exchanges of energy and relative energy exchanges as … (*I think this is important to introduce the terms here otherwise, many readers may be lost depending on their own usage of the terminology*). Energy transfers can occur ...

We appreciate this suggestion and we have partly incorporated this text into the Method section.

2.3.2. Interpreting the bispectrum

**Page 5, Line 30:** "The former are the so-called difference frequencies, while the latter is referred to as a sum frequency": using "former" and "latter" can be confusing because it is not always to what they refer. I usually tend to avoid these terms. I would write instead: "$f1$ and $f2$ are so-called difference frequencies, while $f3$ is referred to as sum frequency".

We agree with R2 and have clarified the text.

I find the subsection "2.3.2 Interpreting the bispectrum" very practical and useful. Nonetheless, in this subsection, the notions of energy gains and losses are mentioned, while they are defined further in subsection 2.3.4. In my opinion. It would be more logical to define the bispectrum, the calculation of geometries from the bispectrum, the energy exchanges, and then show how to interpret the bispectrum. Again, this organisation should ensure that terms and concepts are defined when they are mentioned for the first time in the manuscript, which is important.

On this occasion we disagree with R2. We prefer to let the text follow the order in which we did the computations and in which we present the figures. During the previous review round we already added a sentence to Section 2.3.1. in which we explain the units of energy transfer. For the Method Section this will suffice. In Section 4.2. we further deal with the exact meaning of "energy exchanges" within a palaeoclimatic context.

**Reference:**

Collis, W.B., White, P.R., Hammon, J.K., 1998. Higher-order spectra: the bispectrum and the trispectrum. Mechanical Systems and Signal Processing 12(3), 375-394.

This reference was already part of our reference list.

We would like to take this opportunity to thank M. Martinez for his constructive second review.

[revised manuscript text omitted]